# Cellular morphodynamics as quantifiers for functional states of resident tissue macrophages *in vivo*

Miriam Schnitzerlein[1,2,3]☯, Eric Greto[4,5,6]☯, Anja Wegner[4,5,6], Anna Möller[4,5,6,7], Oliver Aust[4,5,6,7], Oumaima Ben Brahim[4,5,6], David B. Blumenthal[7], Vasily Zaburdaev[1,2]*‡, Stefan Uderhardt[4,5,6]*‡

**1** Department of Biology, Friedrich-Alexander-Universität Erlangen-Nürnberg (FAU), Erlangen, Germany, **2** Max-Planck-Zentrum für Physik und Medizin, Erlangen, Germany, **3** Max Planck Institute for the Science of Light, Erlangen, Germany, **4** Department of Medicine 3 - Rheumatology and Immunology, FAU and Universitätsklinikum Erlangen, Erlangen, Germany, **5** Deutsches Zentrum für Immuntherapie, FAU, Erlangen, Germany, **6** Exploratory Research Unit, Optical Imaging Competence Center Erlangen, FAU, Erlangen, Germany, **7** Department of Artificial Intelligence in Biomedical Engineering, FAU, Erlangen, Germany

☯ These authors contributed equally as first authors.
‡ These authors contributed equally as corresponding authors.
* stefan.uderhardt@fau.de (SU); vasily.zaburdaev@fau.de (VZ)

**Data availability statement:** All imaging data is available on Zenodo with DOI

## Abstract

Resident tissue macrophages (RTMs) are essential for tissue homeostasis. Their diverse functions, from monitoring interstitial fluids to clearing cellular debris, are accompanied by characteristic morphological changes that reflect their functional status. While current knowledge of macrophage behavior comes primarily from *in vitro* studies, their dynamic behavior *in vivo* is fundamentally different, necessitating a more physiologically relevant approach to their understanding. In this study, we employed intravital imaging to generate dynamic data from peritoneal RTMs in mice under various conditions and developed a comprehensive image processing pipeline to quantify RTM morphodynamics over time, defining human-interpretable cell size and shape features. These features allowed for the quantitative and qualitative differentiation of cell populations in various functional states, including pro- and anti-inflammatory activation and endosomal dysfunction. The study revealed that under steady-state conditions, RTMs exhibit a wide range of morphodynamical phenotypes, constituting a naïve morphospace of behavioral motifs. Upon challenge, morphodynamic patterns changed uniformly at the population level but predominantly within the constraints of this naïve morphospace. Notably, aged animals displayed a markedly shifted naïve morphospace, indicating drastically different behavioral patterns compared to their young counterparts. The developed method also proved valuable in optimizing explanted tissue setups, bringing RTM behavior closer to the physiological native state. Our versatile approach thus provides novel insights into the dynamic behavior of *bona fide* macrophages *in vivo*, enabling the distinction between physiological and pathological cell states and the assessment of functional tissue age on a population level.

10.5281/zenodo.13929787. The scripts to analyse the imaging data is available on GitHub at link https://github.com/MiriamSchnitzerlein/MacrophageMorphodynamics, the DOI to this repository (as issued by Zenodo) is 10.5281/zenodo.10563067.

**Funding:** SU is supported by the DFG (project-IDs 448121430, 405969122, 447268119, https://www.dfg.de), by an ERC starting grant (project-ID 101039438, https://erc.europa.eu/apply-grant/starting-grant) and by the Hightech Agenda Bavaria. VZ acknowledges the support by the DFG (German Research Foundation); project A7 within the Research Training Group "Immunomicrotope" (GRK 2740/447268119). The funding agencies did not play any role in the study design, data collection and analysis, decision to publish or preparation of the manuscript.

**Competing interests:** We have read the journal's policy and the authors of this manuscript have the following competing interests: DBB consults for BioVariance GmbH.

## Author summary

In this study, we combine state-of-the-art *in vivo* imaging with advanced computational analysis to reveal the dynamic behavior of peritoneal resident tissue macrophages (RTMs) in their natural environment. These sentinel cells, which are crucial for tissue homeostasis, constantly monitor their environment and, in the process, undergo dynamic morphological changes that have remained largely uninvestigated due to technical limitations. Using two-photon microscopy, we captured time-lapse images of RTMs in the peritoneal serosa under various experimental conditions. Our customized image processing pipeline allowed a comprehensive assessment of cell morphology and dynamics and provided unprecedented insights into the behavior of RTMs *in vivo*, enabling us to distinguish cell populations in different physiological and pathological states. Our work opens up new avenues for the dynamic *in situ* phenotyping of macrophage functionality in disease contexts without their extraction from tissues and provides a novel perspective on the behavior of RTMs in their natural microenvironment. This versatile tool promises to advance our understanding of tissue homeostasis and macrophage function in health and disease, with potential applications in both basic research and clinical settings.

## Introduction

Resident tissue macrophages (RTMs) are the most prevalent immune cells found in healthy mammalian tissues [1], performing both tissue-specific and universal core functions crucial for maintaining homeostasis. Their roles range from synaptic pruning in the brain to surfactant turnover in the lungs and erythrocyte disposal in the spleen [2–4]. Universally, and niche-independently, RTMs surveil interstitial fluids [5], contain acute damage (*cloaking*) [6], and remove cellular debris [7,8], all of which are essential for regular tissue function. These diverse functions are characterized by dynamic changes in cell morphology, which differ significantly from those observed in cultured bone marrow-derived macrophages (BMDMs). Unlike migratory BMDMs, RTMs are non-migratory, sessile cells firmly anchored to the extracellular matrix, performing their functions locally in the interstitial space around them. This unique positioning results in distinct morphologies and behaviors adapted to specific physiological requirements and anatomical constraints of the individual niche, which currently cannot be recapitulated in cell culture systems. A particularly characteristic dynamic behaviour of RTMs *in vivo* is the monitoring of interstitial space by pinocytic 'drinking', which manifests itself in the recurring formation of prominent membrane protrusions and the absorption of larger amounts of fluid. We have previously established a qualitative relationship between macrophage morphology and pinocytotic sampling and cloaking [5]. However, a quantitative understanding of RTM morphodynamics *in vivo* is crucial for developing predictive mathematical models based on optical imaging data. Such an approach, combined with advanced profiling techniques, will enable more accurate diagnosis of macrophage dysfunction in a spatial context and provide deeper insights into tissue-level pathological processes.

Quantifying cell shapes is a powerful approach to distill complex morphologies into interpretable metrics, enabling researchers to distinguish cells based on phenotype or activation state, and to gain insights into proliferation patterns and migratory behavior [9–12]. This approach has become increasingly valuable across various biological disciplines, from developmental biology to immunology. Morphology quantification methods can be broadly

categorized into many different approaches [9,13,14], we here follow the classification from [14]. The first category utilizes 'landmark points', which are distinct structural features on a cell. While this method is less applicable to most cell types, it has proven useful for cells with consistent features, such as neurons [15]. The second approach employs geometric representations like signed distance maps, level sets or Fourier descriptors to implicitly describe cell outlines. For instance, Tweedy *et al.* [16] used Fourier descriptors to characterize unique cell shapes of migrating *D. discoideum* in response to chemical stimuli. The third, and most widely applicable method involves measuring a set of human-interpretable 'features', including cell area, perimeter, axis lengths, circularity and solidity [13,14]. Such features can be extracted in a highly automated manner and have been used before to analyze immune cells such as macrophages in various contexts. Notably, the static morphology of microglia, RTMs in the brain, has been extensively studied using automated analysis pipelines [17–19]. These quantitative methods have also been applied to classify macrophage polarization in cell culture systems, using metrics such as cell elongation or eccentricity [20,21].

Studying the biology of RTMs, particularly their dynamic sampling activity, requires a novel approach that goes beyond analysis of static images. While various shape-quantification methods have been used to characterize macrophages, they were mostly not intended to capture the intricate, time-dependent morphological changes of non-migratory cells embedded in a tissue environment. To address this gap, we employed an advanced intravital imaging platform to generate high-resolution, time-lapse data of murine peritoneal RTMs *in vivo*. This approach allowed us to observe these cells under both steady-state and experimentally perturbed conditions, providing a comprehensive view of their dynamic adaptations within a living tissue. We developed a custom image processing pipeline that defines and tracks a set of human-interpretable cell morphological features of sampling activity over time. This provided unprecedented insights into RTM morphodynamics, revealing quantitative signatures that could distinguish between physiological and pathological cell states and infer qualitative morphological changes from the quantitative measurements.

Our morphometric analysis not only advances our understanding of RTM behavior *in situ* but also provides a foundation for developing more physiologically relevant tissue culture systems and explant platforms. By capturing the dynamic nature of RTM sampling and other functions, this work represents a significant step towards bridging the gap between *in vitro* studies and the complex reality of living organisms.

## Results

### The peritoneal serosa: A model platform for studying RTM biology

Resident tissue macrophages (RTMs) in stromal tissues typically form uniform 2D or 3D matrices, and the peritoneal serosa exemplifies this organizational concept with a unique 2D planar arrangement [6]. In this tissue, RTMs are uniformly distributed within the submesothelial space (see Fig 1A for an overview of the tissue), a region approximately 15-20 μm deep, situated directly beneath the cavity-lining mesothelial layer. Here, RTMs are intricately entangled within a network of collagen fibers (see Fig 1B panel 5) and reside among a dense population of fibroblasts, as depicted in Figs 1B panel 3, 1C and 1D. A key characteristic of these RTMs is their non-migratory nature; they are effectively locked in position within the tissue matrix. Despite this stationary status, they exhibit dynamic sampling activity, forming prominent protrusions primarily used for pinocytotic uptake of extracellular fluids [5]. This sampling activity is confined to the immediate surrounding area and occurs almost exclusively in two dimensions, with no significant protrusion dynamics observed along the z-axis (i.e., towards the cavity). In contrast to the *in situ* behavior of RTMs, the fibroblasts in this

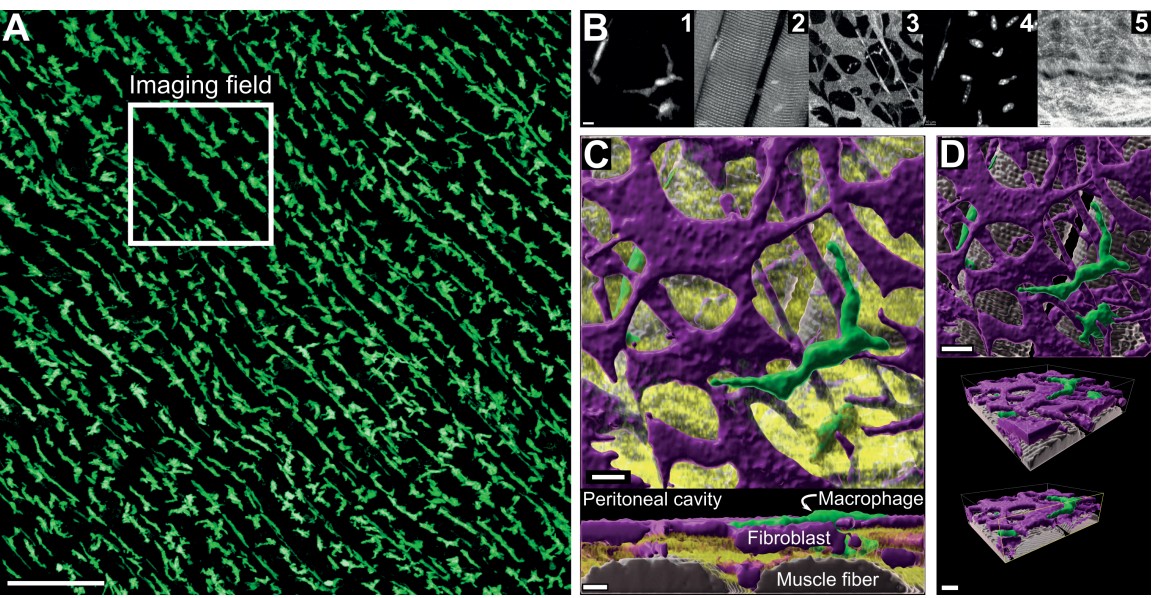

**Fig 1. Volumetric reconstruction of the peritoneal tissue architecture.** (A) Overview microscopy image of a PFA-fixated peritoneum showing uniform distribution of resident macrophages of the peritoneal serosa (green = LysM-tdTomato). Scale bar amounts to 200 μm. (B) Single channels of an isolated volumetric microscopy tile, showing LysM-tdTomato macrophages (1), autofluorescence of muscle fibres (2), CD140+ fibroblasts (3), nuclei stained with Hoechst (4), and 2-photon-generated second harmonics showing collagen fibers (5). Scale is the same for all panels and amounts to 10 μm. (C) Surface reconstruction showing top view (top) and side view (bottom) of the volumetric imaging tile, showing fibroblasts in purple, macrophages in green, muscle fibers in grey, and collagen in yellow. Both scale bars amount to 10 μm. (D) Surface reconstruction of the volumetric imaging tile without collagen, displaying top view (top) and oblique view with slice through the imaging volume. Both scale bars amount to 15 μm.

environment remain stationary also but do not display any notable dynamic behavior, either in steady-state conditions or upon activation [6]. This unique combination of static structural elements and dynamically sampling macrophages makes the peritoneal serosa an ideal prototypical stroma model for studying basic tissue biology. Its relatively simple architecture, free from complex anatomical restrictions, allows for clear observation and analysis of cellular interactions and behaviors. The principles established through studies of the peritoneal serosa can often be translated to other, more complex tissues [6].

## Cell shape features quantitatively describe RTM morphodynamics

RTMs exhibit dynamic sampling behavior crucial to their surveillance function, which manifests as visible changes in cell morphology over time. To characterize this core biological function *in vivo*, we developed a complex approach combining intravital imaging with advanced quantitative analysis.

We acquired three-dimensional, time-lapse images of RTMs in the peritoneal serosa of living mice. Given the planar nature of RTM sampling activity in this compartment [6], we utilized 2D projections for high-throughput analysis, which were corrected for jittering or drifts (see Materials and Methods for details). Representative sample images of a macrophage are shown in Fig 2A, from which cell morphology and dynamics are extracted and analyzed – a corresponding video to this figure can be found in the supplemental S1 Video.

To characterize RTM morphology, we employ intuitive quantifiers that describe both cell size and shape as illustrated in Fig 2B. The parameters offer easier interpretation compared to complex measures like Fourier descriptors. Our custom pipeline extracts and analyzes these

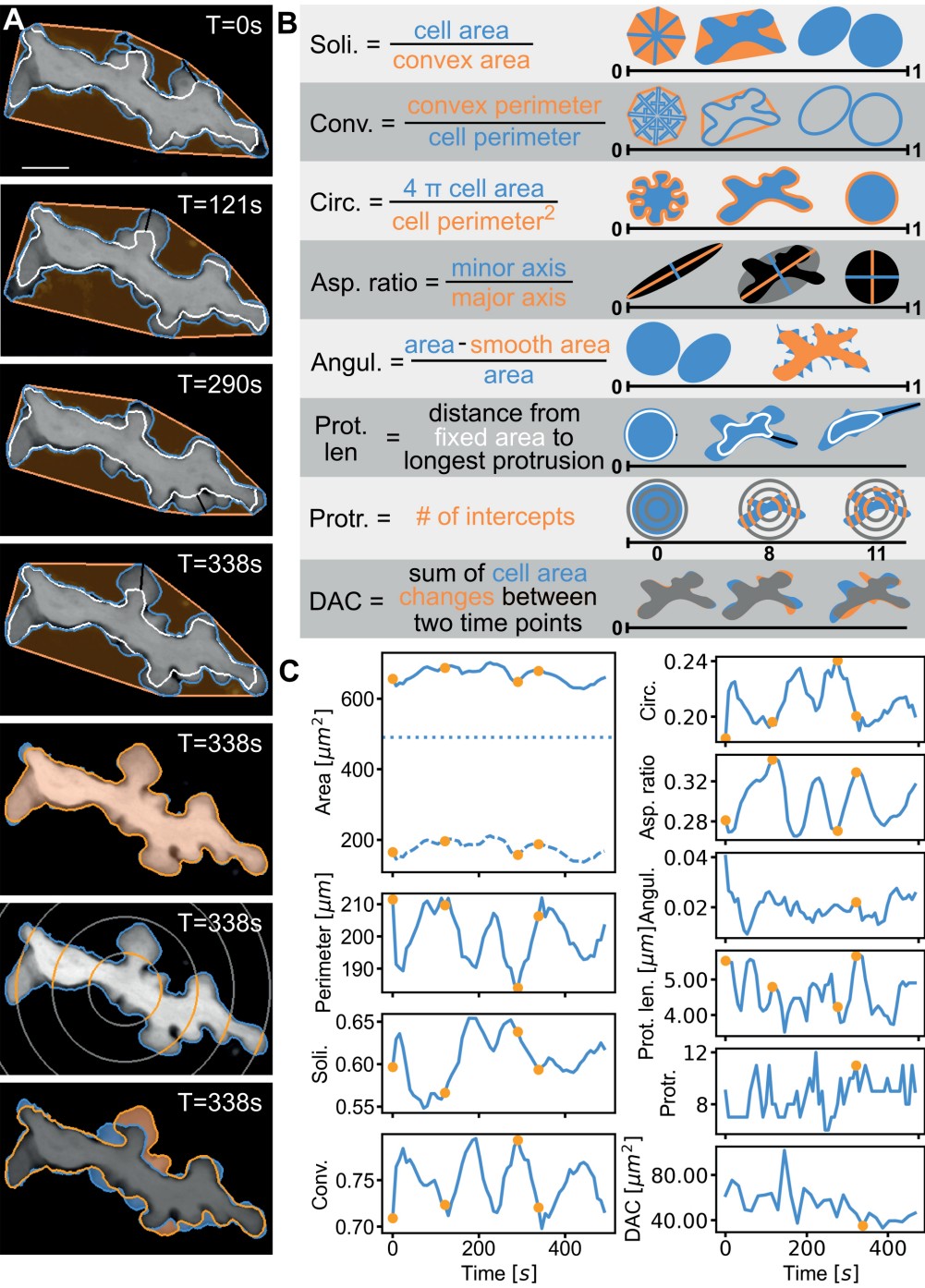

**Fig 2. Visualization of cell size and cell shape for a single exemplary RTM over time.** (A) Snapshots of an RTM at representative time points with the outline of the whole cell marked in cornflowerblue. Panels 1-4: additionally marked is the outline of fixed area (white), outline of convex hull (orange) and length of the longest protrusion of the cell (black). Panel 5: outline of the cell (cornflowerblue) with smoothed cell boundary (orange). Panel 6: concentric Sholl shells (grey) for protrusiveness, their intercepts with the cell body marked in orange. Panel 7: cell at time 338 seconds (outline in cornflowerblue) plotted together with the same cell at time 353 seconds (outline in orange). Their overlayed and thus shared area is colored in gray. The scale bar equates to 10 μm. (B) Definition of cell shape parameters and corresponding sketches qualitatively visualizing potential cell shapes. (C) Cell size and shape of the exemplary cell from (A) over time. On the left side from top to bottom: Whole cell area (drawn through), (constant) fixed cell area (dotted) and mobile cell area (dashed); perimeter; solidity; convexity. On the right, from top to bottom: circularity, aspect ratio, angularity, length of the longest protrusion, protrusiveness, dynamic area change. The time points of the movie snapshots in (A) are marked with orange dots on the time series.

features as detailed in the Methods section. Cell size is quantified via the 2D-projected area and perimeter. We further differentiate between fixed and mobile areas, reflecting the sessile nature of RTMs [6]. The fixed area is determined by integrating the movie and applying a 97% threshold, while the mobile area represents the difference between total and fixed area. The evolution of these parameters over time is plotted for a representative example cell in Fig 2C.

We selected eight shape parameters:

1. Solidity: Ratio of cell area to convex area, measuring cell spread.
2. Convexity: Ratio of convex perimeter to cell perimeter, evaluating outline "wrinkliness".
3. Circularity: Assesses roundness, calculated from area and perimeter ratio.
4. Aspect ratio: Ratio of minor to major axis, quantifying elongation.
5. Angularity: Measures surface irregularities using erosion-dilation operations [22].
6. Longest protrusion length: Maximum distance between cell boundary and fixed area outline.
7. Protrusiveness: Adapted Sholl analysis [23,24], counting intersections with concentric circles with multiples of 8 μm radii. While this was originally intended as a quantifier for dendritic organisation of neurons, this feature was already successfully used to characterize macrophage morphology [24].
8. Dynamic area change (DAC): Area change over a period of 14 seconds (14 seconds is the smallest temporal resolution available for some experiments and is thus chosen as lowest common denominator). For that the cell at those two time points is compared and the cell area which is not superimposed is summed up.

All of these cell shape quantifiers - except the protrusiveness, protrusion length and DAC - range from 0 to 1 and are qualitatively explained for different shapes in Fig 2B. These quantifiers are easily interpretable and thus enable us to make statements about the overall cell shape changes by tracking these values as a function of time, see Fig 2C. Notably, several parameters, including mobile/fixed areas, protrusion length, and DAC, can only be determined from dynamic data, underscoring the importance of analyzing RTM dynamics for understanding their behavior in various states.

The cell shown in Fig 2A, actively sampling at rest, showed no long-term trends, indicating that our imaging protocol and processing pipeline effectively mitigated potential artifacts, including photo-toxicity or photo-bleaching, resulting in consistent segmentation results and feature extraction. We found that the changes of the cell size and shape quantifiers (Fig 2C) were fluctuating only within a relatively narrow range (e.g., $\leq 15\%$ for the perimeter), while the overall cell morphology was visibly highly dynamic (compare panels 1 and 2 of Fig 2A). These discrepancies could be explained by the expansion (Fig 2A, panel 2 and 4) and retraction (Fig 2A, panel 1 and 3) of membrane protrusions, which occurred almost cyclically and inversely. This periodic sampling behavior was captured in several shape parameters over time, such as the convexity (Fig 2C, bottom left).

Having validated our approach for quantifying RTM equilibrium states, we applied these features to discriminate between RTMs exposed to different chemical stimuli, representing various functional states. To achieve this, we summarized each features' time series into a discrete value, enabling time-independent characterization of macrophage morphology and dynamics.

## Morphodynamical quantifiers discriminate between RTMs in physiological and pathological cell states

To facilitate comparison of RTMs across different states, we introduced summary statistics that reduce time-dependent measurements to single parameter values per cell. These statistics capture three key aspects of cell behavior:

1. Average cell morphology: Represented by the mean of each feature.
2. RTM dynamics: Assessed via the standard deviation of features, normalized by their respective means to account for size-dependent variations. Additional dynamic quantifiers include:
   - Ratio of mobile to fixed area, indicating the cells sampling potential.
   - Slope of the cumulative sum of Dynamic Area Change (DAC), quantifying the rate of shape change.
3. Active adaptation: Measured as the slope of a linear fit to the time series, indicating ongoing morphological polarization or response to new stimuli.

To assess the sensitivity of different quantification methods, we exposed resident tissue macrophages to various *in vivo* stimuli known to induce either inflammatory or pro-resolution polarization states, each associated with distinct molecular and transcriptional signatures. Given the mostly short-term nature of our stimulation protocol, we did not assess transcriptional changes. However, confocal immunostaining of fixed tissues revealed molecular responses characterized by the nuclear translocation of key transcription factors. These observations reveal response patterns that parallel polarization trajectories documented in *ex vivo* and *in vitro* studies, exemplified by the pronounced nuclear translocation of p65 following LPS stimulation (as visualised in Fig 3 and Table 1) [25]. We used the following stimuli:

- "M-CSF": Macrophage Colony Stimulating Factor (M-CSF) serves as a crucial differentiation and survival factor for macrophage precursors. It activates the PI3K/Akt signaling pathway and significantly enhances the pinocytotic activity of macrophages, among its diverse biological functions [5].
- "TGF-$\beta$": Transforming Growth Factor $\beta$ (TGF-$\beta$) plays a crucial role in tissue development and homeostasis while acting as an anti-inflammatory and pro-resolution mediator, partially through activation of the transcription factor STAT3 [26,27].
- "LPS": Lipopolysaccharides (LPS), components of the Gram-negative bacterial cell wall, trigger macrophages to mount an acute pro-inflammatory response through activation of the NF-$\kappa$B signaling pathway, particularly via the p65 subunit [28,29].
- "IFN-$\gamma$": Interferon $\gamma$ (IFN-$\gamma$) is a potent activator of macrophages through the STAT1 signaling pathway inducing antimicrobial and antitumor mechanisms including increased pinocytosis and receptor-mediated phagocytosis [30].
- "YM": YM201636 (YM) is an inhibitor of the kinase PIKfyve which abrogates endosomal resolution and consequently shuts down sampling dynamics [5,31].

Having shown in Fig 3 and Table 1 that RTMs *in vivo* change their functional state upon chemical stimulation - as demonstrated by the localization of relevant transcription factors to the nucleus -, our goal next is to connect the morphodynamic signature of RTMs to their function. Fig 4A shows representative snapshots of RTMs under these conditions. While visual inspection alone was insufficient to discern distinct morphodynamical states, our quantitative analysis was able to reveal significant differences.

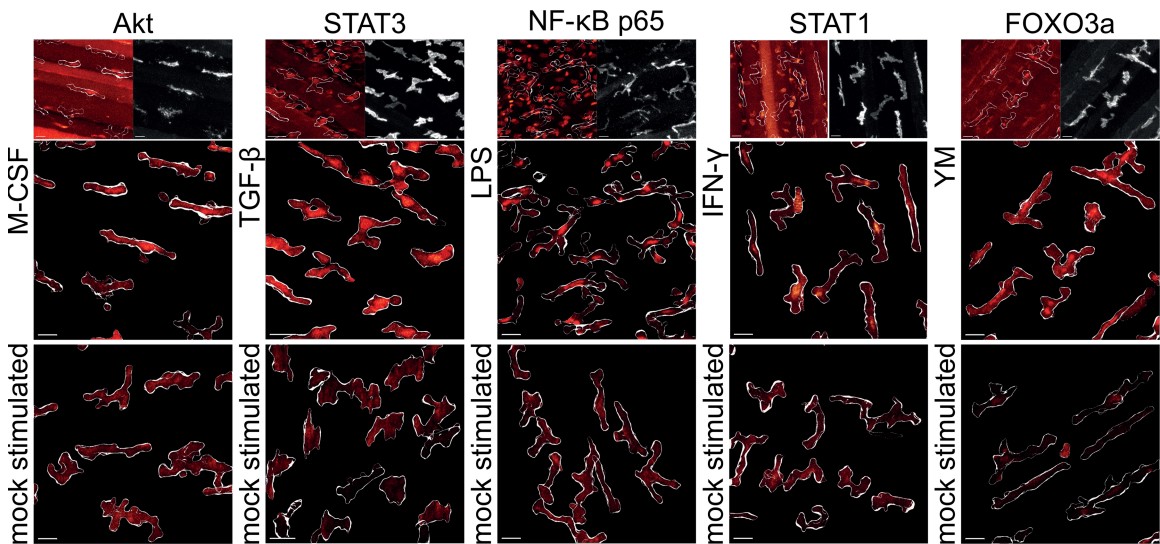

**Fig 3. Comparison of translocation of transcription factors to the nucleus.** Representative confocal microscopy images depicting segmented macrophages and the subcellular localization of transcription factors following stimulation with M-CSF, TGF-$\beta$, LPS, IFN-$\gamma$ or YM201636 compared to mock-treated controls (bottom panels). The top panel displays whole-tissue projections of the respective transcription factor signals (left) alongside CD169 immunostaining (right) used for macrophage segmentation. Scale bar: 20 μm.

**Table 1. Nuclear translocation of transcription factors in response to various stimuli.** Qualitative assessment of transcription factor nuclear translocation following treatment with M-CSF, TGF-$\beta$, LPS, IFN-$\gamma$ or YM201636. Individual macrophages and their nuclei were computationally segmented to visualize nuclear localization of transcription factors. Nuclear translocation was categorized into three distinct patterns: negative [−] indicating no detectable nuclear accumulation, weakly positive [(+)] showing partial nuclear accumulation, and positive [+] demonstrating strong nuclear accumulation with distinct nuclear immunostaining.

| | Nuclear translocation of | | | | |
|---|---|---|---|---|---|
| | Akt | STAT3 | p65 | STAT1 | FOXO3a |
| **M-CSF** | + | (+) | - | - | - |
| **TGF-$\beta$** | (+) | + | (+) | - | - |
| **LPS** | - | (+) | + | - | (+) |
| **IFN-$\gamma$** | - | - | - | + | - |
| **YM** | - | - | - | - | + |

All parameters were analyzed for control cells (Ctrl, N=124) and stimulated cells (M-CSF (N=57), TGF-$\beta$ (N=53), LPS (N=32), IFN-$\gamma$ (N=45) and YM (N=29)). Detailed comparisons can be found in S2 Fig, with key examples shown in Figs 4B-E. We found that certain cell morphology parameters, such as the mean perimeter or the standard deviation of the solidity, were highly suitable to differentiate between different conditions (Fig 4B and 4D). Others, like the mean circularity were sensitive to only specific conditions (Fig 4C). Certain parameters, e.g. the trend of the protrusion lengths (Fig 4E), proved unsuitable for distinguishing between the different *in vivo* conditions.

Our analysis demonstrated that this set of human-interpretable features could successfully differentiate between RTMs under various experimental challenges. Next, we took advantage of selecting human-interpretable features to infer qualitative changes in cell morphology and dynamics from quantitative measurements.

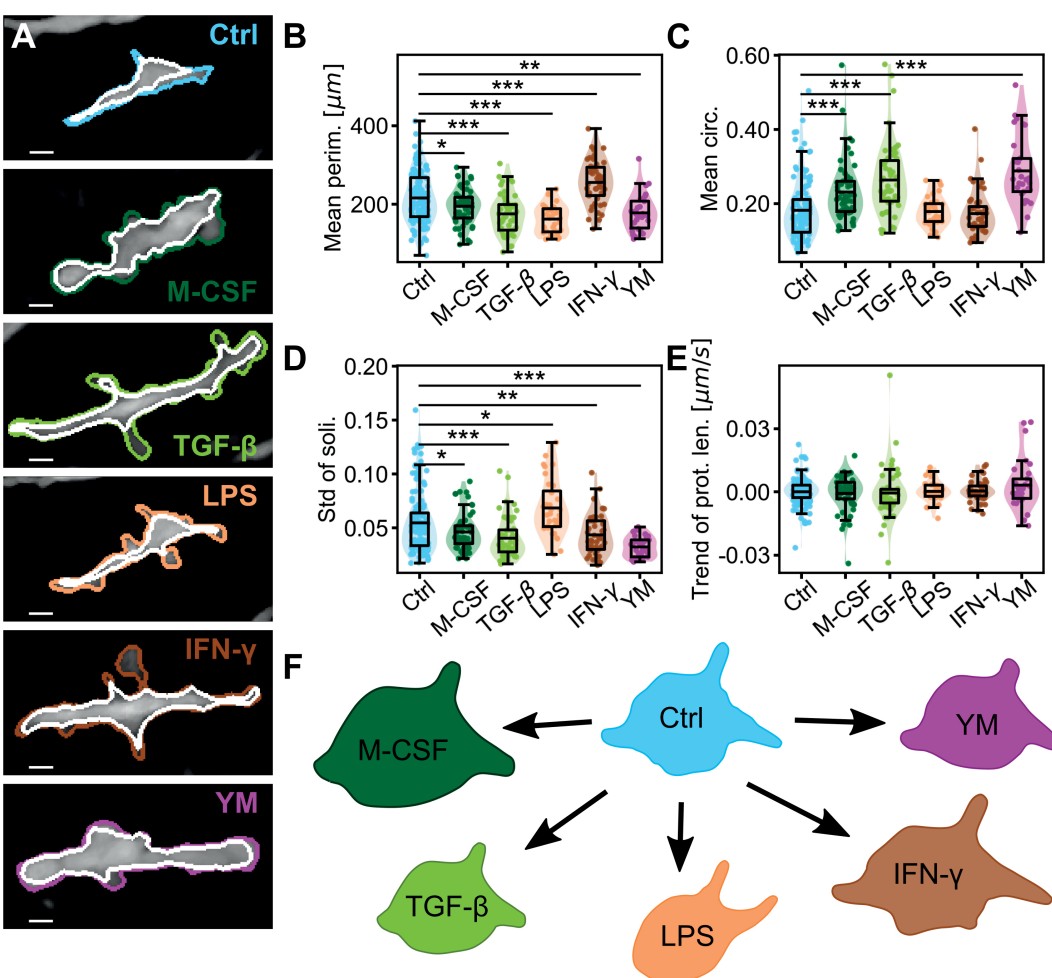

**Fig 4. Comparison of RTMs *in vivo* subjected to different chemical stimulations.** (A) Snapshots of RTMs under different conditions: Control (Ctrl, N=124), Macrophage Colony Stimulating Factor (M-CSF, N=57), Transforming Growth Factor $\beta$ (TGF-$\beta$, N=53), Lipopolysaccharides (LPS, N=32), Interferon $\gamma$ (IFN-$\gamma$, N=45), YM201636 (YM, N=29). The cell perimeter (colored) and the outline of its fixed area (white) are marked; the length of all scale bars is 10 μm. (B-E) Comparison of the different chemical conditions using selected cell size and shape features. The mean of the population is marked, tests for statistical significance were performed using a two-sided permutation Welch's t-test. Significance is abbreviated as * p≤0.05, ** p≤0.01, *** p≤0.001. (B) Mean cell perimeter. (C) Mean circularity. (D) Standard deviation of the solidity. (E) Trend of the maximum protrusion length. (F) Sketch showing (strongly exaggerated) the qualitative changes in RTM morphology for the different chemical stimuli.

## Qualitative morphodynamical changes of RTMs can be deduced from quantitative measurements

Our quantitative measurements enabled the deduction of overall qualitative cell morphology and dynamics changes for RTM populations under various conditions. Detailed plots of all morphodynamic changes can be found in S2 Fig, forming the basis for the following qualitative analysis.

RTMs activated with **M-CSF** exhibited increased size and roundness, evidenced by increased whole and mobile areas, decreased perimeter (Fig 4B), and increased solidity and circularity (Fig 4C). The smoothing of cell boundaries and retraction of protrusions were indicated by decreased convexity, protrusiveness, angularity, and protrusion length. Cell

dynamics appeared reduced, as shown by decreased standard deviations of cell area, solidity, and aspect ratio, as well as a reduced ratio of mobile to fixed area.

**TGF**-$\beta$ exposure resulted in similar morphological changes as M-CSF, with cells rounding up and showing smoother boundaries. The cells appeared less active in changing their morphology, as indicated by decreased standard deviations of cell area and solidity, and a reduced ratio of mobile to fixed area.

**LPS**-treated RTMs showed an overall size reduction, with decreased mean area and perimeter (Fig 4B). The cells exhibited smoother boundaries (decreased convexity and protrusiveness) and a more elongated morphology (decreased aspect ratio). Interestingly, while the ratio of mobile to fixed area and standard deviation of solidity increased, suggesting increased sampling activity, the decreased standard deviations of angularity and protrusiveness hint at fewer but larger protrusions.

**IFN**-$\gamma$ stimulation led to a significant increase in cell size, evidenced by increased cell area and perimeter (Fig 4B). While shape parameters remained largely unchanged, sampling activity appeared strongly decreased, as shown by reduced ratio of mobile to fixed area and standard deviations of various parameters. This aligns with IFN-$\gamma$'s known effect of increasing extracellular fluid uptake.

**YM201636** treatment resulted in rounder, swollen, and less active cells. This was supported by increased fixed area, decreased mobile area, decreased mean cell perimeter (Fig 4B), and increased mean solidity, circularity (Fig 4C), and convexity. The cells showed less complex boundaries and reduced dynamic activity (see Fig 4D), consistent with the expected inability to dissolve endosomes and form sampling protrusions.

Fig 4F summarizes the overall qualitative morphological changes under different conditions.

Thus, we could use our 31 morphodynamic quantifiers to interpret and verify the qualitative cell morphological changes of RTMs under controlled perturbations. Furthermore, we connected their morphodynamic changes due to stimulation to functional states as identified by the translocation of relevant transcription factors to the nucleus. Next, we aimed to facilitate differentiation of cells of different conditions by means of dimensionality reduction of the large set of quantifiers given.

## Dimensionality reduction reveals extensive RTM morphospace and stimulus-specific constraints

Dimensionality reduction of our numerous cell morphology parameters offers a valuable approach to identify key differentiators of RTM morphologies under various conditions. This reduced dimensionality space can potentially assign cells in unknown states to specific conditions or functional states based solely on morphology and dynamics. Interestingly, Principal Component Analysis (PCA) did not yield clear clustering of single cell populations for our given features and conditions (Fig 5A). Other dimensionality reduction methods, including t-SNE and UMAP, produced similar results (S3A-S3D Fig) showing very heterogeneous, intermingled cell populations.

We attribute this lack of distinct clustering to the broad distribution of the control population in the parameter space (the RTM morphospace), which is already evident in Figs 4B-E and S2. This broad distribution, representing substantial heterogeneity in steady-state cell morphology and dynamics, persists across various dimensionality reduction methods (see S3 Fig). Notably, we confirmed that this dispersal was not due to varying experimental conditions in the control populations (Fig 5B). Rather, a comparable wide distribution was found in all control experiments, suggesting that under steady state conditions there is a pronounced

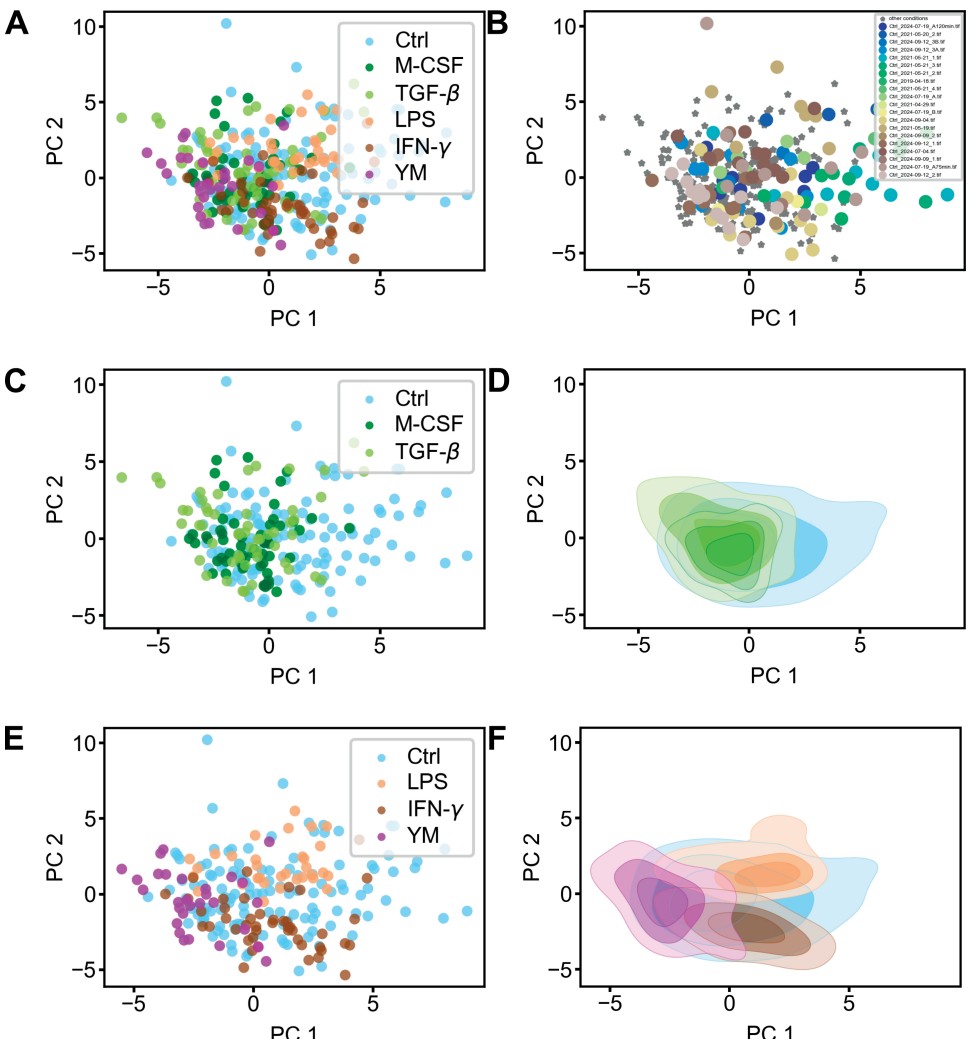

**Fig 5. Dimensionality reduction of RTM features subjected to different chemical stimulations *in vivo*.** (A) PCA using all available conditions, namely unperturbed control (light blue), M-CSF (dark green), TGF-$\beta$ (yellowgreen), LPS (orange), IFN-$\gamma$ (brown) and YM (magenta). (B) PCA of all populations, but with the Ctrl cells colored according to their experiment, while all other conditions (M-CSF, TGF-$\beta$, LPS, IFN-$\gamma$ and YM) are marked as grey stars. (C, E) Using the principal components from (A), but plotting only selected conditions. (C) Plotting only the Ctrl, M-CSF and TGF-$\beta$ cell populations from the PCA from (A). (E) Plotting only the Ctrl, LPS and IFN-$\gamma$ cell populations from the PCA from (A). (D, F) Using the principal components as in (C, E), but plotting the data as density.

heterogeneity in RTM morphodynamics. This heterogeneity makes a classification of single cells to a certain functional or activated state quite impossible, as the steady-state control cells adopt such a wide variety of morphodynamic traits.

The chosen stimulant concentrations aimed to elicit physiological responses rather than extreme reactions, as evidenced by the fact that most stimulated cells reside within the morphospace spanned by the control population (Figs 5C and 5E). Notably, YM-stimulated cells showed multiple outliers, suggesting more drastic changes in cell morphodynamics, potentially indicating progress towards a more extreme, non-functional state.

We hypothesize that while the control condition encompasses cells with widely varying shape and dynamic properties, specific stimulation might narrow this distribution and shift

the population towards one distinct spectrum of the RTM morphospace. To test this, we separated the PCA plot to show selected populations of stimulated cells: pro-homeostatic agents (M-CSF and TGF-$\beta$, Fig 5C) and inflammatory or activity-inhibiting agents (LPS, IFN-$\gamma$ and YM, Fig 5E). Indeed, Fig 5E revealed a narrowing of cell populations and obvious shift in opposite directions of the morphospace, with minimal overlap between LPS- and IFN-$\gamma$-treated cells. This shift is demonstrated even more clearly when estimating the density from Figs 5C and 5E as shown in Figs 5D and 5F, where the shift of populations is especially visible for the inflammatory or activity-inhibiting agents (Fig 5F). This suggests that RTMs adopt a subset of morphodynamic features when subjected to inflammatory or sampling-arresting stimuli. This narrowing was also evident in other dimensionality reduction methods, such as Linear Discriminant Analysis (S3A Fig). Conversely, pro-homeostatic agents did not cause such marked narrowing, with resulting populations slightly shifted while remaining comparable in heterogeneity to the control population.

To add quantification to those qualitative assertions, we used a statistical test to compare two multivariate distributions as introduced by Rosenbaum [32]. Using this test, we compared all six available cell populations in the 31-dimensional space with each other, with the results shown in Table 2. We found that almost all cell populations appear to come from different distributions, showing that the applied stimuli had a shifting or narrowing effect on the stimulated RTM populations, away from the naïve morphospace. Notable exceptions are the populations of cells stimulated with M-CSF and TGF-$\beta$, as both anti-inflammatory cytokines appear to have a similar effect on RTMs. This was already visible in the previous section, where the qualitative morphological changes in the RTM morphodynamics were highly similar, which can also be inferred qualitatively from the density plots of the two populations (Fig 5D). Furthermore, RTM populations stimulated with M-CSF or IFN-$\gamma$ appear to have a similar distribution. As both stimuli lead to an increase in pinocytic uptake of RTMs, similar morphodynamic changes can be expected. Finally, the population of IFN-$\gamma$ stimulated cells could not be distinguished from the unperturbed RTM population, hinting that stimulation with IFN-$\gamma$ largely retains the heterogeneity of macrophages in the spanned morphospace.

PCA analysis allows easy linking of principal components (PCs) to original parameters. PC1 primarily comprised static cell features integrated over time (mean perimeter, mean solidity, mean circularity or mean protrusiveness), while PC2 was influenced by truly dynamic features (like the standard deviation of the perimeter or of the solidity). Trend quantifiers, aspect ratio, and angularity had minimal effect on PCs. However, trend measurements remain valuable for determining active adaptation to a changing environment and the respective adaptation timescales. The consistency of PCs across different condition sets (S3E and S3F Figs) suggested that the control population primarily determined PC composition.

**Table 2. P-values of the comparisons of cell populations in the 31-dimensional feature space using Rosenbaums test for multivariate distributions. Significance is abbreviated as * $p \leq 0.05$, ** $p \leq 0.01$, *** $p \leq 0.001$.**

| Condition | Ctrl | M-CSF | TGF-$\beta$ | LPS | IFN-$\gamma$ | YM |
|---|---|---|---|---|---|---|
| Ctrl | - | | | | | |
| M-CSF | ** | - | | | | |
| TGF-$\beta$ | ** | 0.157 | - | | | |
| LPS | *** | *** | *** | - | | |
| IFN-$\gamma$ | 0.077 | 0.166 | ** | *** | - | |
| YM | *** | ** | *** | *** | *** | - |

In conclusion, we found that steady-state RTMs exhibit a broad distribution of cell shape and dynamics, spanning an extensive morphospace of physiological shapes and dynamic motifs. Inflammatory stimulation constrained the morphodynamical phenotype of RTMs within this space, leading to a shift of the morphodynamic characteristics of the cell population which we quantified using the Rosenbaum test. We next aimed to apply these quantification factors in a clinically relevant context, specifically to analyze aging-related changes in RTM morphodynamics.

## M-CSF reverses ageing-related morphodynamic alterations in RTMs

Aging is known to induce various changes in mammalian bodies, including homeostatic imbalance which can lead to a chronic low-grade pro-inflammatory state [33]. To quantify aging-related effects on peritoneal macrophage morphodynamics, we imaged and analyzed RTMs from one-year-old mice using our pipeline. Detailed comparisons of all parameters are available in S4 Fig, with key quantifiers shown in Fig 6B–6D.

Our analysis revealed that aged RTMs were smaller and rounder, as evidenced by decreased cell area and perimeter, increased solidity and circularity, and reduced maximum protrusion length (Fig 6C). Their boundaries appeared smoother, indicated by decreased angularity and increased convexity. Notably, the dynamics of aged RTMs were significantly reduced, as shown by decreased standard deviations of multiple parameters (particularly cell perimeter, solidity, convexity, circularity (Fig 6B), and aspect ratio) and a lower ratio of mobile to fixed area. These altered morphodynamic features shifted the aged RTM population partially outside the morphospace of young, healthy control cells in the PCA (Figs 6E and 6F).

We then investigated whether these aging-induced morphodynamic changes could be reversed by stimulation with the pro-homeostatic macrophage growth factor M-CSF. Remarkably, M-CSF treatment rendered the shape and size of aged RTMs largely indistinguishable from young control RTMs, with only a slightly increased mean convexity. The dynamics of the control population were also largely recovered, with standard deviations of most parameters (except protrusiveness) returning to the range of the control population. However, clear trends in morphodynamic quantifiers were observed, differentiating the 'Old+M-CSF' RTMs from their young counterparts and suggesting ongoing adaptation to M-CSF exposure, e.g., through transcriptional regulation. The PCA visualization confirmed this recovery, showing the treated aged population shifting back into the morphospace of the control population (Fig 6E), which is shown even more clearly in the density estimation of the given PCA (Fig 6F). This recovery is also quantitatively shown using the Rosenbaum test, in which the 'Old+M-CSF' cell population could not be distinguished from the control population (p=0.056). In contrast, the population of aged RTMs could be distinguished from both the control population (p=$2.2 \cdot 10^{-4}$) and the 'Old+M-CSF' population (p=$2.5 \cdot 10^{-5}$).

In conclusion, our cell morphodynamic analysis effectively demonstrated the behavioral differences between aged and young RTMs under steady state conditions. Importantly, we showed that M-CSF stimulation could largely restore the morphodynamics of aged RTMs to match those of young, healthy cells. This approach provides valuable insights into age-related changes in RTM function and potential interventions.

To further increase the utility of our analytical approach, we have finally applied our RTM morphodynamic quantifiers to a new experimental context: the establishment of a practical explant platform for imaging the peritoneal serosa. This platform was designed for the study of RTM morphology and dynamics *in situ*, yet outside of the living organism.

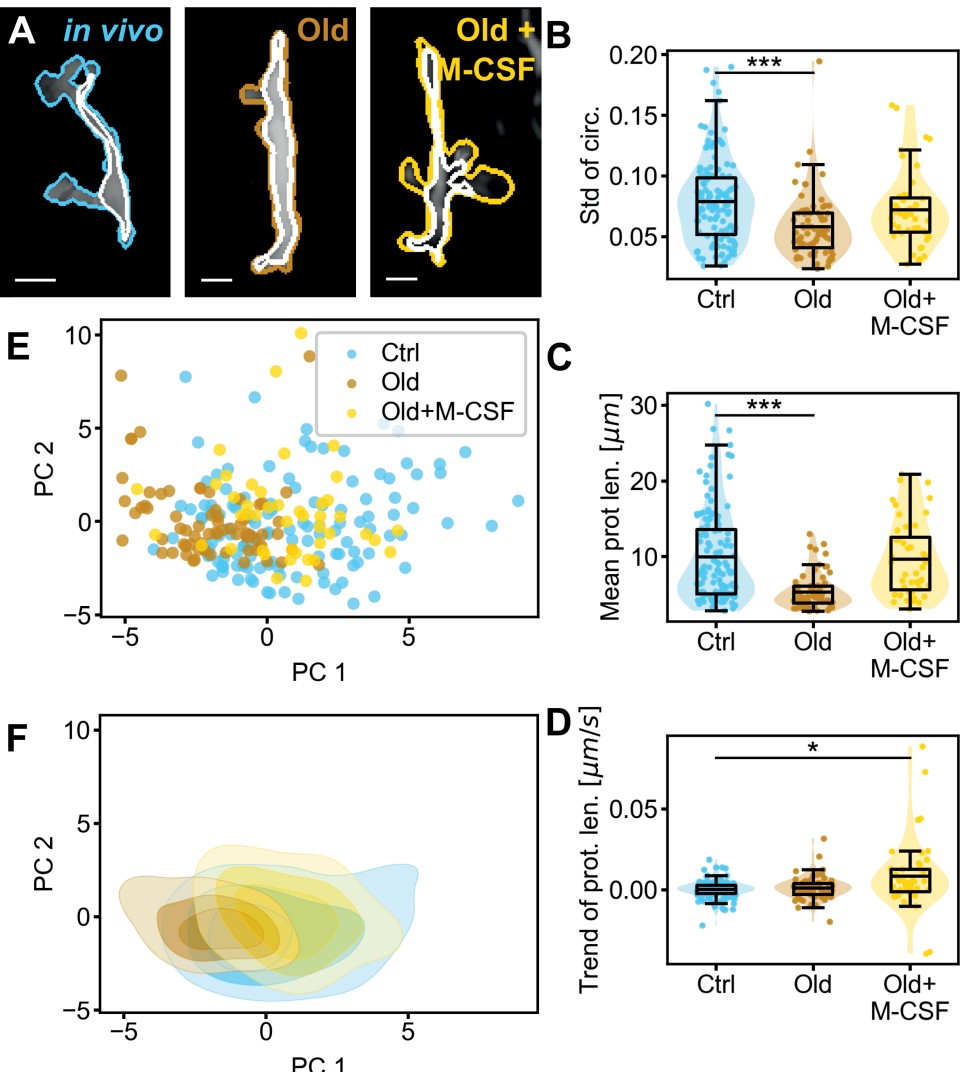

**Fig 6. Comparison of RTMs of young versus aged mice *in vivo*.** (A) Snapshots of RTMs from mice of different ages and with different chemical stimulation. Peritoneal macrophages from young mice serve as the control population (N=124), which are compared to peritoneal RTMs from 1 year old mice without (Old, N=70) and with M-CSF stimulation (Old+M-CSF, N=44). Scale bars amount to 10 μm. (B-D) Comparison of the different populations using key selected cell size and shape parameters - for an extensive comparison please see S5 Fig. The mean of the population is marked, tests for statistical significance were performed using a two-sided permutation Welch's t-test for population comparison. Significance is abbreviated as * $p \leq 0.05$, ** $p \leq 0.01$, *** $p \leq 0.001$. (B) Standard deviation of circularity. (C) Mean of the maximum protrusion lengths. (D) Trend of the maximal protrusion length. (E) Principal component analysis (PCA) of all cell size and shape parameters for the three cell populations. (F) Density estimation of the PCA of the cell populations plotted in (E).

## Improving tissue explant fidelity through quantitative analysis of RTM morphodynamics

Intravital imaging is crucial for understanding immunological processes in living organisms [34]. However, animal experiments are complex and susceptible to various external influences that can affect the quality of the experiment and the downstream analysability (e.g. respiratory movements, effects of prolonged anaesthesia, disturbed fluid balance,

hypoxia). Furthermore, the necessary setup often does not allow for more precise experimental applications, such as the use of micromanipulators. These challenges necessitate more accessible and practical alternatives. Explanting an organ or tissue offers better control over experimental parameters, enhancing versatility. However, it is critical to ensure that the explanted conditions closely mimic *in vivo* conditions. Our shape analysis pipeline serves as an ideal instrument to monitor and verify the comparability of cell shapes and dynamics between explant and intravital imaging conditions.

To investigate this possibility, we explanted mouse peritoneal tissue attached to the abdominal wall, and transferred it to a culture system for subsequent imaging, containing a distinct medium composition widely used for various tissue culture applications (50% MEM, 24% HBSS, 25% FBS, 1% L-Glutamin; Expl 1, N=50) [35]. Visual assessment of snapshots or movies proved insufficient to evaluate the health of explanted RTMs (see Fig 7A). Instead, we applied our human-interpretable cell morphodynamic quantifiers to assess specific differences between RTMs *in vivo* and in explanted tissues (S6 Fig).

Our analysis revealed that explanted RTMs (Expl 1) had smaller mobile areas and cell perimeters. Their cell shape was smoother and less spread-out, evidenced by increased mean solidity (Fig 7B), convexity, and circularity. Decreased mean protrusiveness and maximum protrusion length suggested retracted protrusions. The dynamics of explanted RTMs were also reduced, as indicated by decreased standard deviations of various parameters (see Fig 7C) and a lower ratio of mobile to fixed cell area.

These observations suggest the conditions in the explant imaging setup, while sufficient for cell survival and general sampling behavior, could be improved upon with the goal of moving closer to *in vivo* macrophage dynamics. We therefore adopted a new medium mix based on electrophysiological studies with intact mouse brain slices [36] (Expl 2, N=58): 95% mACSF, 5% FBS, and 50 µM ascorbic acid. While the parameters for cell size and shape remained in close proximity to those of Expl 1, the dynamic state of the cells was largely restored. The standard deviations of the previously diminished parameters showed no significant changes compared to the *in vivo* control population (Fig 7C), and no trends in cell morphodynamics were observed (Fig 7D).

The PCA of all available cell morphology parameters revealed high heterogeneity among the three experimental conditions (Fig 7E). Again, this heterogeneity was not due to variability in individual experiments, as cells from the same experiments scattered rather than clustered together (S7E and S7F Figs). Other dimensionality reduction methods (e.g. t-SNE, UMAP) confirmed this heterogeneity S7A-S7D Fig. Furthermore, the composition of PC1 and PC2 described above remained valid for the PCA analysis of the explanted cell populations.

Our analysis also underscored the importance of examining both static and dynamic parameters of cell morphology. Focusing solely on static measures could have led to the erroneous conclusion that the second explant condition (Expl 2) was unsuccessful, as mean cell size and shape appeared similar between the two explant populations. However, the critical distinction between the two experimental setups emerged only when considering dynamic parameters, such as standard deviations and trends in the data. These dynamic measures revealed significant improvements in the second explant conditions, highlighting the value of comprehensive morphodynamic analysis in evaluating cellular behavior and experimental conditions.

Comparing the explanted RTM cell populations in the 31-dimensional feature space using Rosenbaums test showed a shift of the explanted RTM populations ($p \leq 0.0001$ for both) compared to the control population, which again underscores the need for further improvements of the explantation platform.

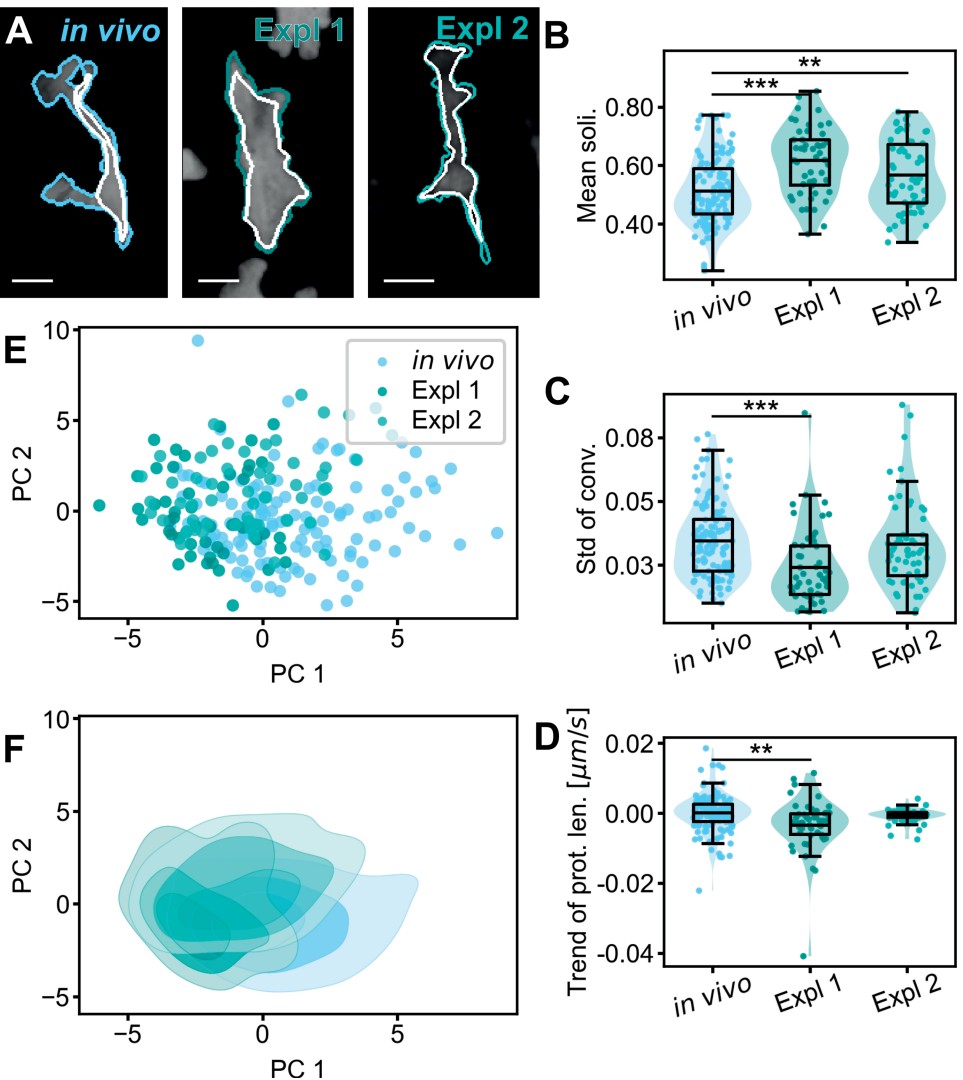

**Fig 7. Comparison of RTMs *in vivo* to explanted RTMs.** (A) Snapshots of RTMs with different experimental setups. Peritoneal macrophages imaged *in vivo* serve as the control population (N=124). The tissue was explanted and the tissues were either submersed in cell culture media (Expl 1, N=50) or a media mixture predominately composed of artificial cerebrospinal fluid (Expl 2, N=58). Scale bar amounts to 10 μm. (B-D) Comparison of the different populations using key selected cell size and shape parameters - for an extensive comparison please see S6 Fig. The mean of the population is marked, tests for statistical significance were performed using a two-sided permutation Welch's t-test for population comparison. Significance is abbreviated as * p≤0.05, ** p≤0.01, *** p≤0.001. (B) Mean solidity. (C) Standard deviation of convexity. (D) Trend of the maximal protrusion length. (E) Principal component analysis (PCA) of all cell size and shape parameters for the three cell populations. (F) Density estimation of the PCA shown in (E).

In conclusion, our improved explantation method largely preserved RTM dynamics compared to *in vivo* conditions and maintained the heterogeneity of cell morphology and dynamics. However, significant differences in cell size and shape persist, indicating that further optimization of the apparatus setup and culture conditions is needed to better match the morphology of explanted RTMs to their *in vivo* counterparts. This approach demonstrates

the power of our morphodynamic analysis in developing and refining *ex vivo* experimental systems to more closely mimic *in vivo* conditions.

## Discussion

In this study, we developed a novel approach to quantitatively analyze the morphodynamics of resident tissue macrophages (RTMs) of the peritoneal serosa in their native environment. This approach generated unique insights into RTM biology and opened new avenues for understanding tissue homeostasis and immune responses.

Our pipeline parameters quantitatively captured the qualitative behavior and morphological changes of RTMs over time in a human-interpretable manner. We employed these quantifiers to distinguish cell populations in various functional states - including those activated by inflammatory stimuli - from their resting equilibrium state. The quantitative measurements allowed direct inference of qualitative morphological changes associated with these functional states. Additionally, we demonstrated how RTM morphodynamics shifted with age, deviating from the behavior of young macrophages, and subsequently showed that M-CSF treatment restored morphodynamics to those of young, naive cells. Lastly, we applied cell morphology features to assess RTM health in explanted tissues, enabling us to propose improvements to the experimental protocol.

The approach presented here relies on the accurate segmentation of individual cells over time to quantitatively measure their morphology and dynamics. We utilized projected 2D images for segmentation, as cellular dynamics almost exclusively occurred along the *x*- and *y*-axes of the imaging plane. Despite the loss of third-dimension information, we observed no significant long-term trends in individual cell morphology features, suggesting volume conservation by RTMs. However, our segmentation approach has limitations. Some cells were excluded from analysis due to unreliable detection of cell body parts, particularly thin protrusions with low signal intensities. The process is not fully automated, requiring manual adjustments and estimation of thresholds, such as determining fixed cell areas. Additionally, our current approach failed to separate cells overlapping in the *z* dimension, reducing the number of analyzable cells. To address these challenges, improved algorithms for disentangling 3D cell overlap are urgently needed. Artificial intelligence applications, such as Cellpose [37], may enhance reliable detection of entire RTM cell bodies, though in our hands the currently existing segmentation approaches regularly fail to detect protrusions even with extensive training efforts. However, our pipeline can be readily applied to improved segmentation results, potentially expanding its capabilities and accuracy.

The repertoire of RTM shape and dynamics quantifiers can be further expanded. Cell outline curvature analysis, previously applied to arteries, could be adapted for cells like RTMs [38]. Tracking cell contour displacement would enable identification of active and inactive membrane areas, providing insights into protrusion dynamics and movement patterns. Temporal analysis of sampling behavior could be enhanced by integrating quantifiers over time, using techniques such as Fourier transforms to detect periodicity or autocorrelation functions to identify characteristic time scales. This approach may reveal oscillatory patterns, as partially visible in Fig 2C, offering deeper insights into RTM sampling mechanisms at both cellular and tissue levels. An expanded parameter set could enable PCA-based grouping and automated classification of RTMs, potentially predicting their activation status with quantifiable confidence. Conversely, this enhanced analysis may also uncover greater heterogeneity and diversity among cells within a given tissue niche than previously recognized.

Based on the processing pipeline and quantifiers used in this study, we found that RTMs within a seemingly uniform anatomical compartment, exhibit remarkable heterogeneity in

their morphodynamics, even under steady-state conditions [6]. We used PCA to create a physiological RTM morphospace that represents the heterogeneous diversity of morphodynamic motifs. This diversity could be an expression of local micro-environmental imprinting [39], or may reflect the adaptability of these cells to respond rapidly to environmental changes and stressors [40]. The observed variability could also indicate cyclical patterns of activity, where periods of active sampling alternate with phases of internalized material processing. Further investigation is needed to elucidate the temporal dynamics and potential periodicity of these behaviors. Generally, our approach has potential applications for studying RTMs across diverse tissue niches and organs. While the stromal environment shapes macrophages' transcriptional phenotype, tissue-specific factors likely also influence their morphodynamics. However, current technical limitations impede the study of macrophage morphodynamics in other peripheral tissues with the spatial and temporal resolution and versatility achieved in our peritoneal platform.

Our pipeline's application across various experimental conditions demonstrated its effectiveness in distinguishing peritoneal RTM functional states. We quantified and characterized morphological changes induced by diverse stimuli, including pro-inflammatory agents and homeostatic factors. Inflammatory activation significantly reduced heterogeneity and narrowed the range of observed morphodynamic motifs, though all behaviors remained within the naïve morphospace boundaries. This suggests that RTMs activated with LPS, for instance, are directed towards specific dynamic functions associated with particular inflammatory responses, as evidenced by the clear distinction between LPS- and IFN-$\gamma$-stimulated populations. In contrast, pro-homeostatic stimulation with TGF-$\beta$ or M-CSF maintained a broad distribution of morphodynamic motifs, with slight shifts along the PC1 axis. *In vivo*, M-CSF primarily induces macropinocytosis, promoting the uptake of fewer but larger fluid volumes [5]. This aligns with our observations of enlarged RTMs and reduced micropinocytotic uptake activity that is facilitated through increased dynamic with multiple protrusions. Our analysis of aged RTMs revealed significant alterations in their morphodynamics compared to young cells. Notably, these changes extended beyond the "young morphospace", creating a distinct, only partially overlapping "aged morphospace" markedly shifted along PC1. Such a distinct "aged morphospace" would be expected when considering the context of inflammaging - the notion that with aging a chronic low-grade inflammation state is developed [33]. The ability of M-CSF treatment to restore a more youthful phenotype by shifting the aged morphospace back within the young morphospace boundaries suggests a potential decline in the M-CSF/M-CSF-receptor axis in aging tissues. On another note, with aging, tissues often acquire varying proportions of monocyte-derived macrophages alongside the prenatally-derived RTMs, which may exhibit distinct morphodynamical behaviors. This possibility was not addressed in our current study, as specific reporter systems, such as the Ms4a3-tdTomato mouse model are required to reliably distinguish between these populations [41]. Nevertheless, this finding highlights the potential for targeted interventions to modulate RTM function in age-related contexts.

This approach proved additionally valuable in optimizing *ex vivo* experimental setups. By quantifying RTM morphodynamics in explanted tissues, we assessed how closely these conditions mimic the *in vivo* state. Through informed improvements to tissue culture conditions, we enhanced dynamic behavior patterns in our custom explant setup. We propose cellular morphodynamics as a crucial indicator of physiological tissue state, complementing methods like transcriptional phenotyping. This directly quantifiable target should be considered when developing and optimizing *ex vivo* or *in vitro* model systems. Our modular analysis pipeline now provides researchers with a powerful tool for direct, quantitative evaluation of

new explant methods, tissue culture systems, and platform designs. This enables precise comparisons between the morphodynamics of explanted RTMs and their *in vivo* counterparts.

## Materials and methods

### Ethics statement

All animal experiments were conducted in accordance with German guidelines and laws and were approved by the responsible local animal ethics committees of the government of Mittelfranken - namely the "Regierung von Unterfranken - Veterinärwesen, Verbraucherschutz" located in Würzburg (study protocol 55.2.2-2532-1172). The animals were housed and bred in the animal facilities of the Friedrich-Alexander-University Erlangen-Nuremberg under specific-pathogen-free conditions.

### Intravital imaging of the peritoneal serosa

For intravital microscopy of the peritoneal serosa LysM-Cre x tdTom F1 animals between 10 to 18 weeks have been used to allow the visualization of peritoneal macrophages. Intravital imaging was done as previously [6] described with slight modifications. Imaging was performed on a Zeiss LSM 880 NLO two photon microscope equipped with a 20x water immersion objective (Zeiss; W PlanApochromat 20x/1,0 DIC D=0,17 M27 75 m m, Cat#421452-9880-000). Animals were anesthetized with isoflurane (cp pharma; Cat#1214). The animal was shaved and a midline incision was made into the abdominal wall. Subsequently, another incision into the peritoneal wall was made along the linea alba to expose the peritoneal serosa. The peritoneal wall was carefully retained using two small sutures (Ethicon; coated VICRYL-polyglactin 6/0; Cat#V134H) and mounted on a $12 \times 25$ mm plastic base plate with a raised 12 mm circular plate using Vetbond (3M; Cat#1469SB). 10 μL of sterile pre-warmed HBSS buffer with Ca+ and Mg+ (Gibco; Cat#14025-050), containing M-CSF or LPS, was applied onto the surface, before covering the serosa with a sterilized cover glass (Paul Marienfeld; Cat#4663433). The animal was then transferred to the heated microscopic imaging chamber. The serosa was kept at a temperature of 36 ° C using an adjustable heater (WPI; Air-Therm SMT). Animals which showed signs of surgical damage or bleeding were not studied further. Data were acquired at a resolution of $512 \times 512$ (12 bit) in stacks of 12 frames each 3 μm apart at a frame rate of 943.72 ms and a zoom of 1.5 using ZEN software (Zen 2.1 SP3; Zeiss). For imaging, the laser was tuned to 930 nm and the following filter cubes were used: (SP 485; BP 500-550; BP 575-610).

### *In vivo* stimulation of RTMs

Chemical stimulation was performed as previously described [5]. In brief, acute stimulation (100 ng/mL M-CSF, 100 ng/mL LPS) was performed by adding the stimulating agent to 10 μL of sterile pre-warmed HBSS buffer with Ca+ and Mg+ (Gibco; Cat#14025-050). This was directly applied to the peritoneal serosa surface and covered with a sterilized cover glass (Paul Marienfeld; Cat#4663433). Imaging was conducted after a 30-minute incubation period. For endosome resolution blocking, tissues were first incubated with 100 nM YM201636 for 1 hour, followed by the addition of 100 ng/mL M-CSF to enhance macropinocytotic uptake. Imaging was performed after an additional 30-minute incubation. Notably, we used a significantly lower concentration of YM201636 to decrease, rather than completely inhibit, RTM dynamics, as higher concentrations cause cells to cease sampling and round up [5].

For TGF-$\beta$ or IFN-$\gamma$ stimulation, mice were injected intraperitoneally with 0.2 mg/kg TGF-$\beta$ or 0.5 mg/kg IFN-$\gamma$ dissolved in 200 μL sterile HBSS. Imaging was performed 24 hours post-injection.

## Immunostaining of fixed peritoneal tissues

Tissues were stimulated with LPS, TGF-$\beta$, IFN-$\gamma$, M-CSF or YM201636 as described above and subsequently fixed using formaldehyde fixativ (BD Cytofix/Cytoperm Fixation/Permeabilization Kit; Cat #554714) overnight at 4 ° C. After extensive washing with PBS (Gibco; Cat #10010023), tissues were permeabilized and blocked using a saponin-containing buffer (BD Perm/Wash Perm/Wash Buffer; Cat #554723) with 5 % normal goat serum (Invitrogen; Cat #31872) for at least 1 hour at room temperature. All samples were then stained with Hoechst nuclear dye (dilution 1:1000; BD Pharmingen Hoechst 33342 Solution; Cat #561908) and anti-mouse CD169 Alexa Fluor A488 (clone 3D6.112; dilution 1:200; Biolegend; Cat #142419), and either unconjugated rabbit anti-mouse STAT3 (clone D3Z2G; dilution 1:100; Cell Signaling; Cat #12640), STAT1 (clone D1K9Y; dilution 1:100; Cell Signaling; Cat #14994), FOXO3a (clone D19A7; dilution 1:100; Cell Signaling; Cat #12829), pan-Akt (clone C67E7; dilution 1:100; Cell Signaling; Cat #4691), or NF$\kappa$B p65 (clone D14E12; dilution 1:100; Cell Signaling; Cat #8242) in Perm/Wash buffer overnight at 4 ° C. After extensive washing with Perm/Wash buffer, samples were incubated with goat anti-rabbit Alexa Fluor 647 (polyclonal; dilution 1:2000; Invitrogen; Cat #A-21245) for 3 hours at room temperature. Samples were then washed with Perm/Wash buffer and mounted in Fluoromount-G mounting medium (Invitrogen; Cat #00-4958-02) for confocal imaging. Same microscope was used for static imaging as for live imaging, only with following adjusted acquisition parameters (96-slice z-stacks recorded at 8bit depth, image size 512 x 512 pixels, and pixel scaling of 0.42 μm x 0.42 μm x 1.00 μm. Analysis and visualization was performed using Imaris 10.2.0 (Bitplane).

## Explantation and imaging of the explanted peritoneal serosa

For the explantation of the peritoneal serosa, animals were prepared in a similar way as for the intravital imaging of the serosa. Animals were sacrificed, shaved and a midline laparotomy performed. A chalazion forceps was carefully inserted into the abdomen without touching the peritoneal serosa. After tightening the forceps, holding the serosa in place, scissors were used to cut around the forceps. The tissue was immediately transferred to a vessel with prewarmed media. Media was either 50% FCS, 25% HBSS, 24% MEM, 1% P/S for Expl 1, or 95% mACSF with 5% FCS and 50 μM ascorbic acid for Expl 2. Imaging parameters were kept the same as for the *in vivo* imaging.

## Image segmentation and labeling

As described previously, imaging was performed with low laser power and high detector gain to protect the tissue from phototoxicity and photobleaching which could potentially alter cellular behavior [42]. However, this approach leads to a reduced signal-to-noise ratio. To address this, an image analysis pipeline was employed to denoise the images and enhance the true signal - the single steps of the image processing pipeline can be found in Fig 8.

Raw images were processed via an analysis pipeline designed to enhance the signal from the fluorescent macrophages, remove noise, and determine the boundaries of each cell. All images are 3D videos, otherwise referred to as 3D+T data. The 3D+T raw files were stabilized

with a multi-temporal stabilization algorithm implemented in the Correct 3D Drift [43] plugin for the popular image analysis software Fiji [44]. The stabilized data was then denoised with the deep learning based denoising software DeepCad [45]. As published, DeepCad can only process 2D+T datasets. To accommodate 3D+T data, each video was split into a series of 2D+T slices along the Z-axis. A custom model was trained in DeepCad using selected 2D+T slices which contained visible signal with minimal empty frames. After denoising, the datasets were reassembled along the original axes to reconstruct the 3D+T images.

The software Ilastik was then used to train pixel classification models on subsets of the denoised data, which were subsequently applied to all datasets [46]. Unique models were trained to account for different image resolutions and experimental conditions. Ilastik generated probability maps in which the intensity of each voxel represented the probability of it belonging to the 'signal' class rather than the 'background' class. Following pixel classification, the images were restabilized, if necessary, using the Correct 3D Drift plugin in Fiji. The data were then projected along the Z-axis to create 2D+T images, with the 'Sum of slices' option in Fiji employed during projection to average out residual noise in the probability maps - then all further processing and extraction steps were performed in Python.

The 2D probability maps (Fig 8C) were segmented with the RTMs as foreground using an adaptive threshold via Li's iterative Minimum Cross Entropy Method [47] as implemented in the Python package scikit [48]. In a further preprocessing step before labeling the cells, the objects (i.e. cells) of the resulting binary image (Fig 8D) were smoothed using binary opening as implemented in the scikit-image package [48] with a circular structuring element of radius 0.6 μm. In the following step, small holes in the cells smaller than 10 μm and small objects in the background smaller than 10 μm were removed, resulting in the smoothed binary image as shown in Fig 8E.

In order to label the cells individually the watershed algorithm - also implemented in scikit-image [48] - was used which requires a set of seed labels. These seed labels were generated by applying a threshold of 0.97 to the mean of the movie over time (Fig 8G and 8H), removing any objects that are smaller than 50 μm and then assigning individual labels to the remaining distinct objects (Fig 8H). Using these labeled objects as input for the watershed algorithm to label the cells from Fig 8D, a preliminary labeled image was obtained as shown in Fig 8J. Here, protrusions that were disconnected in the binarization process were not labeled as belonging to the cells, as seen for example for the red cell at the top left of the image. In order to correct for that, the binary image from Fig 8D was dilated using a diamond-shaped structuring element with a radius of 2.2 μm and the preliminary labeled image (Fig 8J) was used as seed labels for the watershed algorithm (Fig 8K and 8L). Finally, the resulting labeled and dilated cells are used as overlay for the smoothed binary image (Fig 8M) and the final labeled image is obtained (Fig 8N). As visible in Fig 8N, some labeled objects might actually be composed of two cells, whereas others are cut off at the border of the frame (see e.g. the blue cell at the top or the yellow cell at the bottom right). Therefore, the labeled movies were individually examined and compared to the 3D images to verify which of the labeled objects are actually individual and whole macrophages. In this step, there were also macrophages identified that were labeled as two distinct objects, which was corrected by merging the initial seed labels that wrongfully had two different labels for one single cell. The number of RTMs labeled correctly itemized by the different conditions using the just described method is summarized in Table 3.

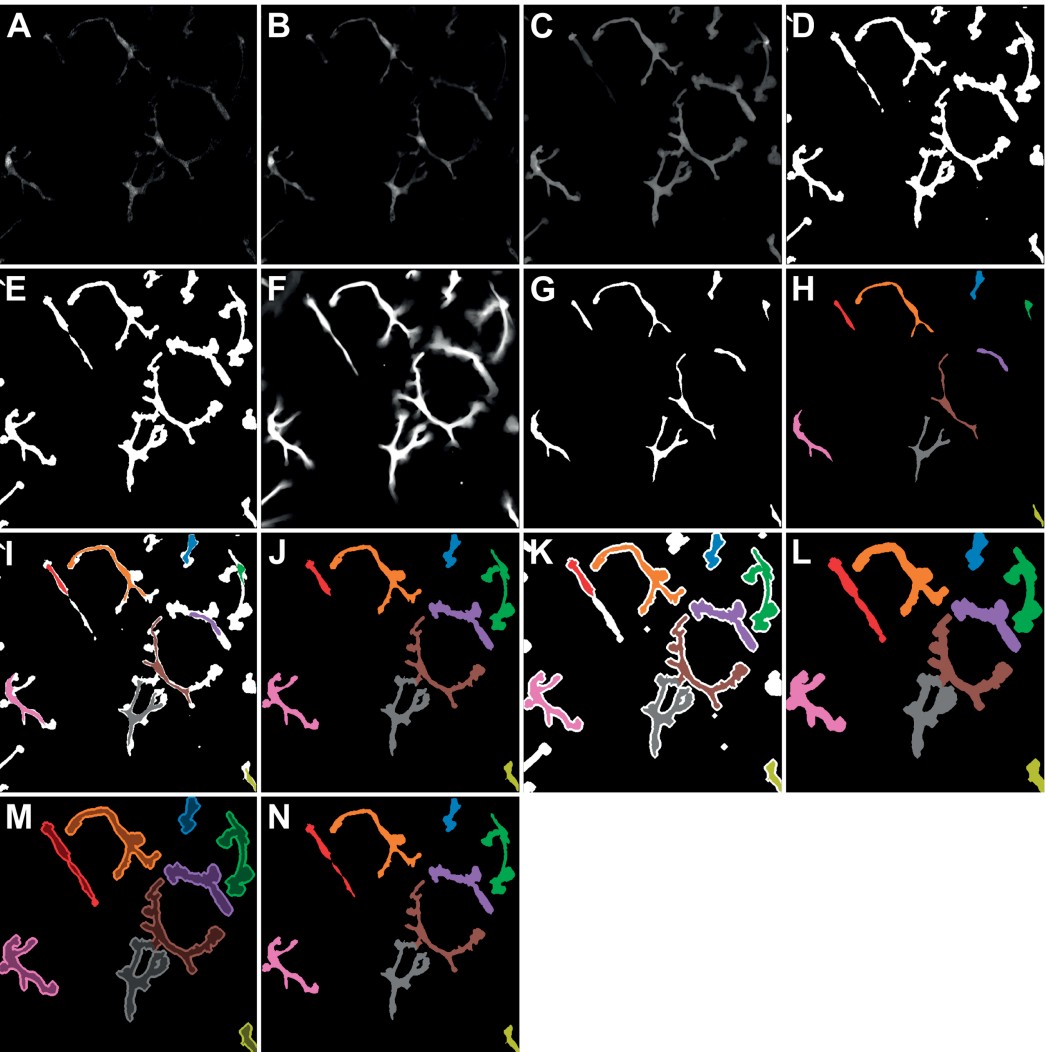

**Fig 8. Visualization of the image segmentation and labeling pipeline for the first frame of an example movie.**
(A) Grayscale raw image. (B) Grayscale image showing the output of DeepCad. (C) Probability map generated by ilastik. (D) Binarized image obtained (from (C)) using thresholding via Li's iterative Minimum Cross Entropy Method. (E) Smoothed binarized image. (F) Integrated movie over all time frames. (G) Fixed area of the integrated movie by using a threshold of 0.97 on (F). (H) Label seeds obtained from (G) by coloring all distinct objects. (I) Overlay of label seeds (H) and binary segmented cells (D). (J) Labeling all distinct objects in (I) using the watershed algorithm to be used as new seed labels. (K) Dilating the cells from (E) and using the labeled cells from (J) as seed labels for the watershed algorithm to achieve the labeling as shown in (L). (M) Using the labels from (L) to correctly label the pre-processed binary images from (E). (N) Final labelled image.

## Comparison of two multivariate populations

To compare whether two multivariate populations might have the same distribution, a statistical test as introduced by Rosenbaum [32] was implemented. For a detailed description of this method the reader is referred to said publication, while it is shortly summarized hereafter for the example of our RTM cell populations. Given two populations of e.g. cells that each

**Table 3. Summary of the extracted amounts of RTMs using the labeling pipeline on the available data.**

| Condition | # experiments | # labeled objects | # correctly labeled RTMs |
|---|---|---|---|
| Ctrl | 20 | 282 | 124 |
| M-CSF | 7 | 105 | 57 |
| TGF-$\beta$ | 6 | 164 | 53 |
| LPS | 5 | 118 | 32 |
| IFN-$\gamma$ | 8 | 157 | 45 |
| YM | 3 | 58 | 28 |
| Old | 6 | 122 | 70 |
| Old+M-CSF | 3 | 65 | 44 |
| Expl 1 | 7 | 156 | 50 |
| Expl 2 | 17 | 154 | 58 |

have multiple features, such that each single cell corresponds to a point in a multidimensional space. The main idea is to calculate the distances between all points from both populations, which are then used to construct an optimal non-bipartite matching such that the result consists of paired observations which minimize the total inter-point distances. Then, the cross-matching statistics, i.e. the amount of pairs made up from points from the two different populations, are used as basis for the test. The idea here is that when the points come from two very different distributions, there should be very few cross-matched pairs while there would be many cross-matches when the points come from two very similar distributions. In this manuscript, the standardized Euclidean distance is used to calculate the inter-point distances. This choice of distance measure has the advantage of being highly interpretable, while having a standardization of the data incorporated.

## Supporting information

**S1 Fig. Mean image intensity over time for the cell visualised in Fig 2.**
(PNG)

**S2 Fig. Comparing different chemically induced activations *in vivo* using cell size quantifiers.** The boxplots show an extensive analysis of RTMs *in vivo* subjected to different chemical stimuli, as also described in Fig 4 - the different stimulants are: Unstimulated (Ctrl, N = 124), Macrophage Colony Stimulating Factor (M-CSF, N = 57), Transforming Growth Factor $\beta$ (TGF-$\beta$, N = 53), Lipopolysaccharides (LPS, N = 32), Interferon $\gamma$ (IFN-$\gamma$, N = 45), YM201636 (YM, N = 29). (A1) Mean of cell area. (A2) Standard deviation of cell area. (A3) Slope of a linear fit to cell area. (B1) Mean of cell perimeter. (B2) Standard deviation of cell perimeter. (B3) Slope of a linear fit to cell perimeter. (C1) Fixed cell area. (C2) Mobile cell area. (C3) Ratio of the mobile to the fixed cell area. (D1) Mean of solidity. (D2) Standard deviation of solidity. (D3) Slope of a linear fit to solidity. (E1) Mean of convexity. (E2) Standard deviation of convexity. (E3) Slope of a linear fit to convexity. (F1) Mean of circularity. (F2) Standard deviation of circularity. (F3) Slope of a linear fit to circularity. (G1) Mean of aspect ratio. (G2) Standard deviation of aspect ratio. (G3) Slope of a linear fit to aspect ratio. (H1) Mean of angularity. (H2) Standard deviation of angularity. (H3) Slope of a linear fit to angularity. (I1) Mean of the maximal protrusion length. (I2) Standard deviation of the maximal protrusion length. (I3) Slope of a linear fit to the maximal protrusion length. (J1) Mean of protrusiveness. (J2) Standard deviation of protrusiveness. (J3) Slope of a linear fit to protrusiveness. (K1) Trend of the dynamic area changes. The mean of the popula-

tion is marked, tests for statistical significance were performed using a two-sided permutation Welch's t-test. Significance is abbreviated as * p≤0.05, ** p≤0.01, *** p≤0.001.
(TIF)

**S3 Fig. Comparing different dimensionality reduction methods for differently chemically stimulated RTM cell populations *in vivo*.** (A)–(F): Colors are blue for Control, dark green for M-CSF, yellowgreen for TGF-$\beta$, orange for LPS, brown for INF-$\gamma$ and magenta for YM201636. (A)–(D): the dimensionality reduction methods were applied on all six cell populations. (A) Linear Discriminant Analysis. (B) Neighborhood Components Analysis. (C) T-distributed Stochastic Neighbor Embedding. (D) Uniform Manifold Approximation and Projection. (E) Principal Components Analysis using only Ctrl, M-CSF and TGF-$\beta$ cell populations. (F) Principal Components Analysis, using only Ctrl, LPS, IFN-$\gamma$ and YM201636 cell populations.
(TIF)

**S4 Fig. Extensive analysis of the effect of aging on the morphodynamics of peritoneal macrophages *in vivo*.** The RTMs of young mice (2–3 weeks) serve as control population (Ctrl, N = 124), while the RTMs of old mice (1 year) are compared either unstimulated (Old, N = 70), or stimulated with M-CSF (Old+M-CSF, N = 44). (A1) Mean of cell area. (A2) Standard deviation of cell area. (A3) Slope of a linear fit to cell area. (B1) Mean of cell perimeter. (B2) Standard deviation of cell perimeter. (B3) Slope of a linear fit to cell perimeter. (C1) Fixed cell area. (C2) Mobile cell area. (C3) Ratio of the mobile to the fixed cell area. (D1) Mean of solidity. (D2) Standard deviation of solidity. (D3) Slope of a linear fit to solidity. (E1) Mean of convexity. (E2) Standard deviation of convexity. (E3) Slope of a linear fit to convexity. (F1) Mean of circularity. (F2) Standard deviation of circularity. (F3) Slope of a linear fit to circularity. (G1) Mean of aspect ratio. (G2) Standard deviation of aspect ratio. (G3) Slope of a linear fit to aspect ratio. (H1) Mean of angularity. (H2) Standard deviation of angularity. (H3) Slope of a linear fit to angularity. (I1) Mean of the maximal protrusion length. (I2) Standard deviation of the maximal protrusion length. (I3) Slope of a linear fit to the maximal protrusion length. (J1) Mean of protrusiveness. (J2) Standard deviation of protrusiveness. (J3) Slope of a linear fit to protrusiveness. (K1) Trend of the dynamic area changes. The mean of the population is marked, tests for statistical significance were performed using a two-sided permutation Welch's t-test. Significance is abbreviated as * p≤0.05, ** p≤0.01, *** p≤0.001.
(TIF)

**S5 Fig. Comparing different dimensionality reduction methods for aged RTM cell populations *in vivo*.** (A)–(D): Colors are blue for Control, dark gold for old RTMs and yellow for old RTMs stimulated with M-CSF. (A) Linear Discriminant Analysis. (B) Neighborhood Components Analysis. (C) T-distributed Stochastic Neighbor Embedding. (D) Uniform Manifold Approximation and Projection. (E) Principal Components Analysis, where the cells from the old population are marked according to their experiment - Ctrl and Old+M-CSF cells are marked as stars. (F) Principal Components Analysis, where the cells from the Old+M-CSF population are marked according to their experiment - Ctrl and Old cells are marked as stars.
(TIF)

**S6 Fig. Comparing different tissue preparation techniques using cell size and shape quantifiers.** The cells imaged *in vivo* serve as control, explanted cells were imaged using two different media (Expl 1 and Expl 2). (A1) Mean of cell area. (A2) Standard deviation of cell area. (A3) Slope of a linear fit to cell area. (B1) Mean of cell perimeter. (B2) Standard deviation of cell perimeter. (B3) Slope of a linear fit to cell perimeter. (C1) Fixed cell area. (C2) Mobile cell area. (C3) Ratio of the mobile to the fixed cell area. (D1) Mean of solidity. (D2) Standard

deviation of solidity. (D3) Slope of a linear fit to solidity. (E1) Mean of convexity. (E2) Standard deviation of convexity. (E3) Slope of a linear fit to convexity. (F1) Mean of circularity. (F2) Standard deviation of circularity. (F3) Slope of a linear fit to circularity. (G1) Mean of aspect ratio. (G2) Standard deviation of aspect ratio. (G3) Slope of a linear fit to aspect ratio. (H1) Mean of angularity. (H2) Standard deviation of angularity. (H3) Slope of a linear fit to angularity. (I1) Mean of the maximal protrusion length. (I2) Standard deviation of the maximal protrusion length. (I3) Slope of a linear fit to the maximal protrusion length. (J1) Mean of protrusiveness. (J2) Standard deviation of protrusiveness. (J3) Slope of a linear fit to protrusiveness. (K1) Trend of the dynamic area changes. The mean of the population is marked, tests for statistical significance were performed using a two-sided permutation Welch's t-test. Significance is abbreviated as * $p \leq 0.05$, ** $p \leq 0.01$, *** $p \leq 0.001$.
(TIF)

**S7 Fig. Comparing different dimensionality reduction methods for cell populations in differently prepared tissues.** (A)–(D): Colors are blue for *in vivo* control, teal/dark greenish for Explant 1, turquoise/light greenish for Explant 2. (A) Linear Discriminant Analysis. (B) Neighborhood Components Analysis. (C) T-distributed Stochastic Neighbor Embedding. (D) Uniform Manifold Approximation and Projection. (E) Principal Components Analysis, where the *in vivo* cell population and the Explant 2 cell populations are marked as stars whereas the Explant 1 cell population is colored according to their corresponding experiment. (F) Principal Components Analysis, where the *in vivo* cell population and the Explant 1 cell populations are marked as stars whereas the Explant 2 cell population is colored according to their corresponding experiment.
(TIF)

**S1 Video. RTM *in vivo* with measured features animated.** Video showing sampling RTM *in vivo* together with the measured morphodynamic features, corresponding to the snapshots in Fig 2.
(MP4)

**S1 Table. Values of the constituents of the PCAs for different populations.** PCAs were performed on different populations (see Figs 5, 6, 7, S3, S5 and S7), which leads to different structures of the principal components. Those different structures are summarized in this .csv table.
(CSV)

**S1 Text. Supplemental text examining the cell morphodynamic quantifiers and their interplay in more detail.**
(PDF)

## Acknowledgments

MS acknowledges the ongoing academic support from the graduate school IMPRS Physics and Medicine throughout her research. Dynamic imaging was performed on a DFG-funded confocal microscope (project-ID 261193037) and a DFG-funded spinning disc microscope (248122450).

## Author contributions

**Conceptualization:** Vasily Zaburdaev, Stefan Uderhardt.

**Data curation:** Miriam Schnitzerlein, Eric Greto, Anja Wegner, Anna Möller, Oumaima Ben Brahim.

**Formal analysis:** Miriam Schnitzerlein, Eric Greto.

**Funding acquisition:** Stefan Uderhardt.

**Investigation:** Miriam Schnitzerlein, Eric Greto, Anja Wegner, Oumaima Ben Brahim.

**Resources:** Stefan Uderhardt.

**Software:** Miriam Schnitzerlein, Eric Greto, Anna Möller, Oliver Aust, David B. Blumenthal.

**Supervision:** David B. Blumenthal, Vasily Zaburdaev, Stefan Uderhardt.

**Validation:** Miriam Schnitzerlein, Eric Greto, Anja Wegner, Anna Möller, David B. Blumenthal, Vasily Zaburdaev, Stefan Uderhardt.

**Visualization:** Miriam Schnitzerlein.

**Writing – original draft:** Miriam Schnitzerlein.

**Writing – review & editing:** Miriam Schnitzerlein, Eric Greto, Anja Wegner, Vasily Zaburdaev, Stefan Uderhardt.

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
