## [Decision Letter · Decision Letter 0]

12 Apr 2024

Dear Prof. Zaburdaev,

Thank you very much for submitting your manuscript "Cell morphology as a quantifier for functional states of resident tissue macrophages" for consideration at PLOS Computational Biology.

As with all papers reviewed by the journal, your manuscript was reviewed by members of the editorial board and by several independent reviewers. In light of the reviews (below this email), we would like to invite the resubmission of a significantly-revised version that takes into account the reviewers' comments.

We cannot make any decision about publication until we have seen the revised manuscript and your response to the reviewers' comments. Your revised manuscript is also likely to be sent to reviewers for further evaluation.

Sincerely,

Philip K Maini

Academic Editor

PLOS Computational Biology

Jason Haugh

Section Editor

PLOS Computational Biology

Reviewer's Responses to Questions

**Comments to the Authors:**

Reviewer #1: In their manuscript titled "Cell Morphology as a Quantifier for Functional States of Resident Tissue Macrophages," Schnitzerlein et al. introduce a variety of quantification parameters for a more detailed analysis of the morphodynamics of tissue-residing macrophages. While inherently interesting, the manuscript does not adequately articulate the significant advancements and benefits of these newly identified parameters in comparison to previously established analytical techniques. Consequently, the novelty of the study seems modest. Prior research, including studies on microglia, has already identified cell characteristics such as branching, protrusion lengths, sampling activity, and cell rounding. In this study, the adjusted analytical parameters for morphodynamics were employed to examine macrophage behavior under chemical or nutritional disturbances in mouse tissues, focusing on how macrophage morphology is altered under severe experimental conditions.

Regrettably, the authors overlook numerous published studies relevant to imaging RTM dynamics in various tissues (e.g., summarized in a recent intravital imaging review by the Lemmermann group) and make several unsubstantiated claims and conclusions about macrophage behavior both in vivo and in vitro.

Major points:

1) The title of the manuscript is somewhat misleading for two reasons: (A) It implies that cell morphology can be utilized to predict the functional states of macrophages, an assertion that is not thoroughly substantiated within the text (refer also to point 2). (B) The phrase "functional states" evokes expectations of insights into biologically significant macrophage activities, such as distinctions between inflammatory vs. anti-inflammatory, alternatively activated vs. non-resolving, phagocytic vs. non-phagocytic, and migratory vs. stationary states. However, the manuscript does not adequately address these functional states. A title that more accurately reflects the content of the manuscript, such as “Refined Analytical Parameters for the Morphodynamic Assessment of Resident Tissue Macrophages,” would be more appropriate.

The authors even demonstrate that treatment with LPS, which induces macrophages into an inflammatory state—a particularly critical functional state—does not result in discernible morphodynamic changes that can be identified by their methods within the analyzed timeframe. This observation further underscores the discrepancy between the expectations set by the title and the findings presented.

2) Although the authors demonstrate that certain artificial interventions can alter macrophage morphology, there is a lack of rigorous evaluation regarding the predictive capabilities of these morphodynamic changes, contrary to what the title and various assertions within the manuscript suggest. For instance, statements like "We used those features to quantitatively differentiate cells in various functional states," and "... helps distinguish between physiological and pathological cell states," imply a level of predictive accuracy that is not sufficiently substantiated by the data.

The functional implications of detecting a "round" macrophage within the tissue are not clearly defined. Does it indicate that the cell is in an environment where the mouse has been overdosed with isoflurane? Does it suggest a deficiency in pinocytosis? Alternatively, it could imply other conditions, such as the cell preparing for division, losing attachment to the extracellular matrix (ECM), or residing in a low-adhesive tissue environment (see work of Maridonneau-Parini, e.g. PMID: 22999511).

3) A critical aspect that remains unexplored in the study is whether the observed effects of the treatments on macrophage morphology are the result of direct cause-effect relationships or are mediated through indirect influences on the extracellular environment. The question arises whether the impact of YM201636 treatment on macrophages could be secondary to its effects on the tissue environment. Similarly, the observed consequences of isoflurane overdose and tissue hypoxia warrant scrutiny to determine if these responses are directly attributable to macrophages or are indirect outcomes of environmental changes.

In this context, it would have been prudent for the authors to also document and present data on the second harmonic generation (SHG) signal from the connective tissue. This additional analysis would ensure that the extracellular matrix (ECM) environment surrounding the analyzed cells is consistent across samples. Previous research by the Lemmermann lab (PMID: 35343899) has demonstrated that macrophage morphodynamics are significantly influenced by their interactions with the ECM.

4) The initial comparison made between BMDMs moving on two-dimensional culture dishes and serosal macrophages within tissue is notably problematic and, to an extent, unfair. In the in vitro setup described, the macrophages are provided with an environment that allows a high degree of freedom, naturally leading to behaviors characteristic of highly migratory cells. Prior research (e.g., PMID: 35343899) has conducted side-by-side comparisons of macrophage morphologies in three-dimensional Matrigel environments versus mouse tissue. These studies have successfully replicated the physiological morphodynamic states of tissue-resident macrophages in three-dimensional setups that offer a lower degree of freedom. As such, Figure 1 is not only misleading but partially incorrect in its conclusions and should be excluded from the manuscript.

Additionally, the analysis in this paper focuses solely on serosal macrophages, representing just one type of RTM. This specific focus should be clearly stated and maintained throughout the manuscript. It is inappropriate to generalize the findings to all types of RTMs, which can vary significantly in their ontogeny and dynamics and include a mix of macrophages, some of which may be more motile bone marrow-derived macrophages.

5) The authors introduce adjusted analytical parameters for evaluating morphodynamic characteristics of cells, yet they do not convincingly demonstrate that these new parameters offer any advantages over the traditional parameters used to describe cell morphology, such as roundness, sphericity, ellipticity, branching, protrusion length, sampling activity, etc. The utility and superiority of these novel parameters remain questionable, as the paper lacks a direct comparison to show that these adjustments provide a clearer or more predictive analysis of cell morphology under various conditions.

One might argue that the traditional morphological parameters could also effectively identify significant morphological transformations, such as those induced by extreme conditions like isoflurane overdose or the chemical inhibition of pinocytosis. Without a thorough comparison that highlights the enhanced sensitivity or specificity of the proposed parameters in detecting subtle or complex morphodynamic changes, it remains unclear why the new parameters should be preferred over the established ones.

Reviewer #2: The authors have performed intravital microscopy of resident tissue macrophages (RTMs) in mice and analyzed their morphodynamics with a custom-made imagine processing pipeline based on Imaris, Fiji and scikit-image in python. Image stacks were acquired with confocal and spinning disc microscopes. Because the RTMs do not move and show shape changes mainly in one plane, the authors used maximal intensity projections and performed a 2D analysis. Shape features such as circularity were extracted for different conditions and PCA was performed, albeit with little success due to the large variability in the wildtype. Similar results were obtained for explants after sufficient waiting time.

In my assessment, this study is very interesting and potentially rewarding, but it is quite unclear what really has been achieved, compare e.g. with Ref. 6 on the same system, which had a much clearer outcome. In the end, the main achievement seems to be that a pipeline has been established for quantitative analysis. The results section is a mixture of method description and phenomenological observations, but does not provide much mechanistic insight. Let’s take the central statement that “Qualitative morphology changes of RTMs can be deduced from quantitative cell shape measurements”. Here it is shown that some pharmacological interventions lead to measurable changes in the shape features. The backdrop here however seems to be the expectations of the authors of what they know should happen under these conditions. At least this is my understanding of the section title after repeated reading. What is obviously missing here is some new or mechanistic finding that has not been described before.

In summary, it is hard to recommend acceptance at the current stage. In my view, this manuscript can be improved considerably in different directions, depending on how the authors want to continue with this project. For example, they could form some mechanistic hypothesis for RTMs and prove this with their pipeline. Why cannot they use the 3D image information to identify periods of medium take-up (which seems to be a major function of these cells, which monitor interstitial fluids) and mechanistically connect this to other changes in the morphodynamics?

Alternatively, they could develop this project further as a methods paper, but then they should go beyond the state-of-the-art with standard software. I do not agree that ML and DL based methods are “emerging” like written in the discussion, they are already standard. I recommend to look into DL methods and to train a deep network on a fraction of the data and then to demonstrate that for the other fraction a functional state can be predicted from the shape features as suggested by the title. Maybe the authors have different ideas how to proceed in the methodological direction, but at the current state of affairs, they seem not to unleash the full potential of the available methods in this field.

Minor comments

The results section is full of literature survey and explanations that are not really results, but should go into e.g. the introduction. The authors should ask themselves what really is a result and what is existing knowledge.

This is especially true for the explanation on shape analysis from pages 4-6, which definitely are not results. Moreover, this list is not very systematic and it is not explained why e.g. Fourier descriptors are not used.

As an example where such a shape analysis led to mechanistic insight, I point to Keren, Kinneret, et al. "Mechanism of shape determination in motile cells." Nature 453.7194 (2008): 475-480. Can something similar be done here for the RTMs?

Fig. 1: a quantitative analysis of the motility data is missing, e.g. plots of position or MSD over time.

The sentence on line 179 is grammatically not complete.

Language is very strong and should be tuned down, e.g. “These analyses provided unprecedented insight into the dynamic behavior of RTMs”. This sentence is hard to justify given that the shape analysis did not give much insight due to strong variability.

Discussion: to what other systems than RTMs could this pipeline be applied in the future?

Reviewer #3: Uploaded as attachment.

**Have the authors made all data and (if applicable) computational code underlying the findings in their manuscript fully available?**

Reviewer #1: Yes

Reviewer #2: Yes

Reviewer #3: Yes

PLOS authors have the option to publish the peer review history of their article (what does this mean?). If published, this will include your full peer review and any attached files.

Reviewer #1: No

Reviewer #2: No

Reviewer #3: No
---

## [Decision Letter · Decision Letter 1]

9 Jan 2025

PCOMPBIOL-D-24-00139R1

Cellular morphodynamics as quantifiers for functional states of resident tissue macrophages in vivo

PLOS Computational Biology

Dear Dr. Zaburdaev,

Thank you for submitting your manuscript to PLOS Computational Biology. After careful consideration, we feel that it has merit but does not fully meet PLOS Computational Biology's publication criteria as it currently stands. Therefore, we invite you to submit a revised version of the manuscript that addresses the points raised during the review process.

Please submit your revised manuscript within 60 days Mar 11 2025 11:59PM. If you will need more time than this to complete your revisions, please reply to this message or contact the journal office at ploscompbiol@plos.org. Please include the following items when submitting your revised manuscript:

We look forward to receiving your revised manuscript.

Kind regards,

Philip K Maini

Academic Editor

PLOS Computational Biology

Jason Haugh

Section Editor

PLOS Computational Biology

**Reviewers' comments:**

Reviewer's Responses to Questions

**Comments to the Authors:**

Reviewer #1: In the revised manuscript, the authors present additional datasets in which they globally treated serosal tissues with various stimuli (e.g., pro-inflammatory stimuli like LPS or IFN-gamma, and anti-inflammatory stimuli such as M-CSF or TGF-beta) before using their analysis pipeline to examine the morphodynamics of resident tissue macrophage (RTM) populations.

With these new datasets, the study further suggests that artificially induced polarization of macrophages into “functional” states correlates with distinct morphodynamic phenotypes at the population level.

However, the study still does not demonstrate that the analysis pipeline can detect, quantify, or identify actual “functional” states of tissue macrophages within a heterogeneous cell population. For instance, it does not allow for the identification of individual inflammatory or anti-inflammatory macrophages in a truly physiological context, such as during a local tissue infection where diverse cell phenotypes would be expected. In their response letter, the authors clarify that their aim was not to predict functional states but rather to quantify macrophage behavior. While useful, this approach falls short of addressing the potential to detect functional states within mixed populations, which would have been a more novel contribution to tissue macrophage biology. Consequently, the manuscript continues to function more as a methods paper, with modest biological novelty. The work feels more like a foundation for training mathematical learning models based on the current datasets. Although the authors now argue for the value of detecting heterogeneity in macrophage populations within untreated, steady-state tissues, I would contend that this finding is largely intuitive and expected. Additionally, the authors have now included data on macrophage populations in aged tissues, showing that these macrophages exhibit different morphodynamic characteristics compared to those in younger tissues. However, the functional implications of these differences remain unexplored.

It is also notable that several paragraphs still imply that the analytical pipeline can detect, quantify, or identify functional states of macrophages within tissue, despite the study not providing evidence of this capability. This starts with the changed title “Cellular morphodynamics as quantifiers for functional states of resident tissue macrophages in vivo” or the suggested title “Dynamic morphological signatures reveal functional states of resident tissue macrophages in vivo” in response to my previous comments (see response letter). It continues with statements such as “These features allowed for the quantitative and qualitative differentiation of cells in various functional states, including pro- and anti-inflammatory activation and endosomal dysfunction.”, “distinction between physiological and pathological cell states and the assessment of functional tissue age” and “distinguish cells in different physiological and pathological states”

For a meaningful biological advance, it would be essential for the authors to link specific morphodynamic states to concrete indicators of macrophage functionality, such as transcriptomic profiles, NF-kappaB activation, IL1-beta production, or increased phagocytic activity.

I cannot judge whether whether the analysis pipeline itself in combination with the descriptive and correlative nature of the current manuscript is sufficient for the scope of Plos Comp Biol. For me, the here provided presentation of novel parameters for morphodynamic cell analysis combined with the analysis pipeline falls into the category of a methods paper.

Reviewer #2: The authors have performed a very substantial revision, as can be seen from the many changes in the markup copy and from the changes in the author list (shift from Zaburdaev to Uderhardt as last author, inclusion of three new authors from the Uderhardt lab). In detail, they have included new experiments, improved the image processing pipeline, removed problematic Figure 1 and the isoflurane data, sharpened their conclusions and clarified the many issues raised by the reviewers. There are now clear biological results and the image processing is put much better into context. In my assessment, the effort was worth it and this manuscript is now very strong. I recommend publication as is.

Reviewer #3: Thanks for the substantial changes to the manuscript, which deal with most of my previous comments and make things clearer. I'm still not convinced by Fig 4 and the discussion around it that justifies the lack of clear separation in the PCA plots (if these metrics are good enough to distinguish macrophages based on activation state then it should be possible to see this). However, the argument that in control conditions macrophages display a broader array of morphologies which are then restricted to a region of morphology space on activation is sufficient justification for me to look past it here.

**Have the authors made all data and (if applicable) computational code underlying the findings in their manuscript fully available?**

Reviewer #1: **No: **The Zenodo server was not functioning when testing the links from the Data availability statement.

Reviewer #2: Yes

Reviewer #3: Yes

PLOS authors have the option to publish the peer review history of their article (what does this mean?). If published, this will include your full peer review and any attached files.

Reviewer #1: No

Reviewer #2: **Yes: **Ulrich Schwarz

Reviewer #3: No

**Figure resubmission:**
---

## [Decision Letter · Decision Letter 2]

9 Apr 2025

Dear Prof. Zaburdaev,

We are pleased to inform you that your manuscript 'Cellular morphodynamics as quantifiers for functional states of resident tissue macrophages in vivo' has been provisionally accepted for publication in PLOS Computational Biology.

Best regards,

Philip K Maini

Academic Editor

PLOS Computational Biology

Jason Haugh

Section Editor

PLOS Computational Biology

Reviewer's Responses to Questions

**Comments to the Authors:**

Reviewer #1: The authors have addressed my raised points and adapted the manuscript accordingly.

**Have the authors made all data and (if applicable) computational code underlying the findings in their manuscript fully available?**

Reviewer #1: None

PLOS authors have the option to publish the peer review history of their article (what does this mean?). If published, this will include your full peer review and any attached files.

Reviewer #1: No

---

## [Editor Report · Acceptance letter]

PCOMPBIOL-D-24-00139R2

Cellular morphodynamics as quantifiers for functional states of resident tissue macrophages in vivo

Dear Dr Zaburdaev,

I am pleased to inform you that your manuscript has been formally accepted for publication in PLOS Computational Biology. Your manuscript is now with our production department and you will be notified of the publication date in due course.

With kind regards,

Anita Estes
